# Seasonal and interannual variability of landfast sea ice in Atka Bay, Weddell Sea, Antarctica

Stefanie Arndt[1], Mario Hoppmann[1], Holger Schmithüsen[1], Alexander D. Fraser[2,3], Marcel Nicolaus[1]

[1]Alfred-Wegener-Institut Helmholtz-Zentrum für Polar- und Meeresforschung, 27570 Bremerhaven, Germany

[2] Institute for Marine and Antarctic Studies, University of Tasmania, Hobart 7001, Tasmania, Australia

[3]Antarctic Climate & Ecosystems Cooperative Research Centre, University of Tasmania, Hobart 7001, Tasmania, Australia

*Correspondence to*: Stefanie Arndt (stefanie.arndt@awi.de)

**Abstract.** Landfast sea ice (fast ice) attached to Antarctic (near-)coastal elements is a critical component of the local physical and ecological systems. Through its direct coupling with the atmosphere and ocean, fast ice properties are also a potential indicator of processes related to a changing climate. However, in-situ fast-ice observations in Antarctica are extremely sparse because of logistical challenges and harsh environmental conditions. Since 2010, a monitoring program observing the seasonal evolution of fast ice in Atka Bay has been conducted as part of the Antarctic Fast Ice Network (AFIN). The bay is located on the north-eastern edge of Ekström Ice Shelf in the eastern Weddell Sea, close to the German wintering station Neumayer III. A number of sampling sites have been regularly revisited each year between annual ice formation and breakup to obtain a continuous record of sea-ice and sub-ice platelet-layer thickness, as well as snow depth and freeboard across the bay.

Here, we present the time series of these measurements over the last nine years. Combining them with observations from the nearby Neumayer III meteorological observatory as well as auxiliary satellite images enables us to relate the seasonal and interannual fast-ice cycle to the factors that influence their evolution.

On average, the annual consolidated fast-ice thickness at the end of the growth season is about two meters, with a loose platelet layer of four meter thickness beneath, and 0.70 meter thick snow on top. Results highlight the predominately seasonal character of the fast-ice regime in Atka Bay without a significant interannual trend in any of the observed variables over the nine-year observation period. Also, no changes are evident when comparing with sporadic measurements in the 1980s and 90s. It is shown that strong easterly winds in the area govern the year-round snow distribution and also trigger the breakup of fast ice in the bay during summer months.

Due to the substantial snow accumulation on the fast ice, a characteristic feature is frequent negative freeboard, associated flooding of the snow/ice interface, and a likely subsequent snow ice formation. The buoyant platelet layer beneath negates the snow weight to some extent, but snow thermodynamics is identified as the main driver of the energy and mass budgets for the fast-ice cover in Atka Bay.

The new knowledge of the seasonal and interannual variability of fast-ice properties from the present study helps to improve our understanding of interactions between atmosphere, fast ice, ocean and ice shelves in one of the key regions of Antarctica, and calls for intensified multi-disciplinary studies in this region.

## 1 Introduction

The highly dynamic pack ice of the open polar oceans is continuously in motion under the influence of winds and ocean currents (Kwok et al., 2017). In contrast, landfast sea ice (short: fast ice) is attached to the coast or associated geographical features, such as for example a shallow seafloor (especially in Arctic regions) or grounded icebergs, and is therefore immobile (JCOMM Expert Team on Sea Ice, 2015). Fast ice is a predominant and characteristic feature of the Arctic (Dammann et al., 2019; Yu et al., 2014) and Antarctic coasts (Fraser et al., 2012), especially in winter. Its seaward edge may vary between just a few meters and several hundred kilometers from where it is attached to, mostly depending on the local topography and coastline morphology. The main processes for fast-ice formation are either in-situ thermodynamic growth, or dynamic thickening and subsequent attachment of ice floes of any age to the shore (Mahoney et al., 2007b).

In the Arctic, coastal regions that are characterized by an extensive fast-ice cover in winter are for example found in the Chukchi Sea and Beaufort Sea (Druckenmiller et al., 2009; Mahoney et al., 2014; Mahoney et al., 2007a), the Canadian Arctic (Galley et al., 2012), the East Siberian and  Laptev Seas (e.g. Selyuzhenok et al., 2017), and the Kara Sea (Olason, 2016). While the fast-ice cover in these regions comes with its own particular impacts on the respective coastal systems, a common feature is that they have undergone substantial changes in recent decades (Yu et al., 2014). These include a reduction of fast-ice area (Divine et al., 2003), later formation and earlier disappearance (Selyuzhenok et al., 2015) and a reduction of thickness (Polyakov et al., 2003).

Along the Antarctic coastline, the fast-ice belt extends even further from the coast (Fraser et al., 2012; Giles et al., 2008) due to the presence of grounded icebergs in much deeper waters of up to several hundred meters (Massom et al., 2001a). Embayments and grounded icebergs provide additional protection against storms and currents, and are often favorable for the formation of a recurrent and persisting fast-ice cover (Giles et al., 2008). Fast-ice around Antarctica is still usually seasonal rather than perennial, and reaches thicknesses of around 2 meters (Jeffries et al., 1998; Leonard et al., 2006), although it may attain greater ages and thicknesses in some regions (Massom et al., 2010). It mostly forms and breaks up annually as a response to various environmental conditions, such as heavy storms (Fraser et al., 2012; Heil, 2006). Its immediate response to both local atmospheric conditions and lower latitude variability of atmospheric and oceanic circulation patterns via the respective teleconnections (Aoki, 2017; Heil, 2006; Mahoney et al., 2007b) make fast ice a sensitive indicator of climate variability and even climate change (Mahoney et al., 2007a; Murphy et al., 1995). Based on the complexity and significance of fast ice in the Antarctic climate system, there is an urgent need for prognostic Antarctic fast ice in regional models, and later in global climate models, to capture its potential major impacts on the global ocean circulation, as developed recently for the Arctic (Lemieux et al., 2016).

Although fast ice only represents a rather small fraction of the overall sea-ice area in Antarctica (Fraser et al., 2012), it may contribute significantly to the overall volume of Antarctic sea ice, especially in spring (Giles et al., 2008). The presence and evolution of Antarctic fast ice is often associated with the formation and persistence of coastal polynyas, regions of particularly high sea-ice production (Fraser et al., 2019; Massom et al., 2001a; Tamura et al., 2016; Tamura et al., 2012) and Antarctic

Bottom Water formation (Tamura et al., 2012; Williams et al., 2008). Also, it forms an important boundary between the Antarctic ice sheet and the pack ice/ocean, for example prolonging the residence times of icebergs (Massom et al., 2003), mechanically stabilizing floating glacier tongues and ice shelves, and delaying their calving (Massom et al., 2010; Massom et al., 2018). Therefore, one particularly interesting aspect of Antarctic fast ice is its interaction with nearby ice shelves, floating seaward extensions of the continental ice sheet that are present along nearly half of Antarctica's coastline. Under specific oceanographic conditions, supercooled Ice Shelf Water favors the formation of floating ice crystals within the water column (Foldvik, 1977; Dieckmann et al., 1986), as opposed to the regular process of sea-ice formation by heat transport from the ocean towards the colder atmosphere. These crystals may be advected out of an ice-shelf cavity and rise to the surface (Hoppmann et al., 2015b; Mahoney et al., 2011; Hughes et al., 2014). They are eventually trapped under a nearby fast-ice cover and may accumulate in a layer reaching several meters in thickness (Gough et al., 2012; Price et al., 2014; Brett et al., 2020). This sub-ice platelet layer has profound consequences for the local sea-ice system, and forms an entirely unique habitat. Thermodynamic growth of the overlying solid fast ice into this layer (by heat conduction from the ocean into the atmosphere) leads to subsequent consolidation, and the resulting incorporated platelet ice may contribute significantly to the local fast-ice mass and energy budgets. This phenomenon has been documented at various locations around Antarctica (Langhorne et al., 2015 and references therein), and where present, is a defining feature of the local coastal system. Refer to Hoppmann et al. (2020) for a comprehensive review of platelet ice.

The effects of fast ice on the exchange processes between ocean and atmosphere are further amplified by the accumulation of snow, as it forms a thick layer over large portions of the Antarctic sea ice (Massom et al., 2001b). However, the snow cover has opposing effects on the energy and mass budgets of sea ice in the region. On the one hand, due to its low thermal conductivity, snow acts as a barrier to heat transfer from sea ice to the atmosphere and effectively reduces ice growth at the bottom (Eicken et al., 1995). On the other hand, snow contributes significantly to sea-ice thickening at the surface through two distinct seasonal processes: snow-ice and superimposed ice formation. In winter/spring, the heavy snow load leads to the depression of the sea-ice surface below water level, causing flooding of the snow/ice interface. The subsequent refreezing of the snow/water mixture forms a salty layer of so-called snow-ice (e.g. Eicken et al., 1994; Jeffries et al., 1998; 2001). In contrast, in spring and summer, surface and internal snowmelt leads to melt water that can refreeze and form fresh superimposed ice, as it percolates through snow and eventually to the snow-ice interface (Haas, 2001; Haas et al., 2001; Kawamura et al., 2004). Both processes contribute significantly to sea-ice growth from the top, and thus to the overall sea-ice mass budget in the Southern Ocean.

Beyond its contribution to the general sea-ice mass and energy budget in the Southern Ocean, fast ice also plays an important role for the ice-associated ecosystem, as it provides a stable habitat for microorganisms (e.g. Günther and Dieckmann, 1999) and serves as a breeding ground for, e.g., Weddell seals and Emperor penguins (Massom et al., 2009).

Fast ice and its properties as described above have been studied around Antarctica for a long time, especially related to logistical work at the summer and overwintering bases close to the coast of the continent. In order to commonly coordinate and facilitate this research, and thus establish an international network of fast-ice monitoring stations around the Antarctic

coastline, the international Antarctic Fast Ice Network (AFIN) was initiated during the International Polar Year (IPY) 2007/2008 (Heil et al., 2011). Active international partners are, e.g., Australia and China working at Davis Station and Zhongshan Station on the eastern rim of Prydz Bay in East Antarctica (Heil, 2006; Lei et al., 2010), New Zealand working out of Scott Base in McMurdo Sound in the Ross Sea (Langhorne et al., 2015, and references therein), Norway at the fast ice in front of Fimbul Ice Shelf at Troll Station (Heil et al., 2011) and Germany in Atka Bay at Neumayer III (Hoppmann, 2015), both in the vicinity of Dronning Maud Land. The regular, AFIN-related monitoring program at Neumayer III started in 2010 in order to fill the observational gap in the Weddell Sea sector.

Here we present a decade of annual in-situ fast-ice observations in Atka Bay, which is one of the longest and most continuous time series within AFIN so far. The main dataset is a record of fast-ice thickness, snow depth, freeboard, and sub-ice platelet-layer thickness that was collected by a number of overwintering teams between 2010 and 2018. In addition to determining the spatio-temporal variability of the fast-ice cover, we co-analyze this data with meteorological observations and satellite imagery in order to determine how snow and platelet ice influence the local fast-ice mass budget. In doing so, we aim to improve our understanding of the interaction between the atmosphere, fast ice, ocean and ice shelves in one of the key regions in Antarctica.

## 2 Study site and measurements

### 2.1 Study site: Atka Bay

The main study area of this paper is Atka Bay, an 18 km-by-25 km embayment in front of the Ekström Ice Shelf located on the coast of Dronning Maud Land in the eastern Weddell Sea, Antarctica, at 70°35'S/ 7°35'W (Figure 1). Atka Bay is flanked towards the east, south and west by the edges of the ice shelf which rise as high as 20 meters above sea level. The cavity geometry of the Ekström Ice Shelf is one of the best known in Antarctica (Smith et al., 2020). Atka Bay is seasonally sea-ice covered, and the water depth ranges between 80 m and 275 m with maximum depth in the central bay (Kipfstuhl, 1991). Since the 1980s, when the first German research station Georg-von-Neumayer Station was established in the region, a variety of measurements has been carried out in the bay. Today's German research station Neumayer III is located at a distance of about 8 km from the bay, where drifting snow regularly forms natural ramps between the sea ice and the ice-shelf surface. Prior investigations of the interactions between ice shelf, sea ice and ocean in the bay and its surroundings have been carried out by Kipfstuhl (1991) and Nicolaus and Grosfeld (2004), as well as more recently by Hoppmann et al. (2015a) and Hoppmann et al. (2015b). Ecosystem studies from the 1990's have been published by Günther and Dieckmann (1999) and Günther and Dieckmann (2001).

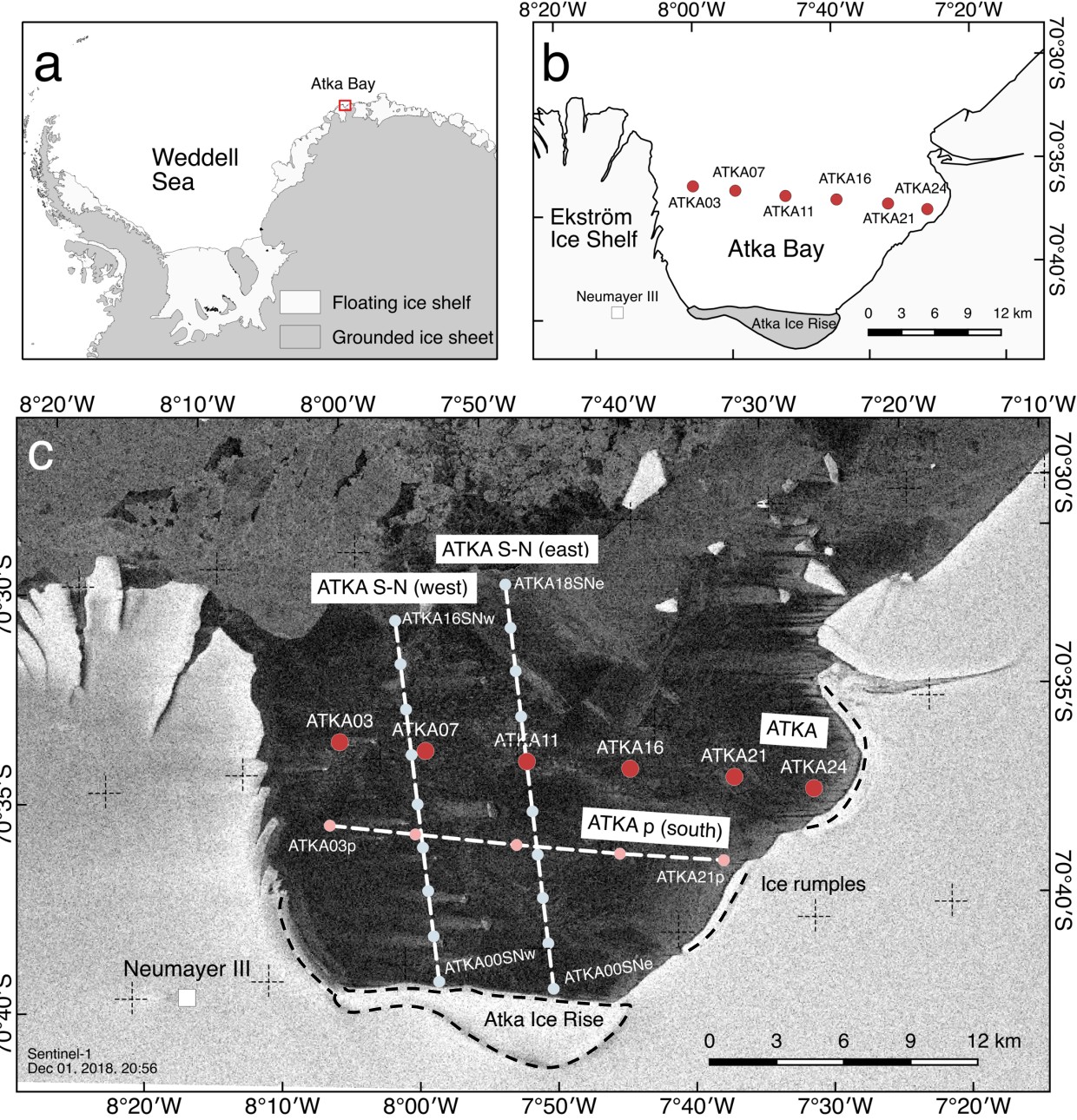

**Figure 1.** Overview of the study site and its surroundings. (a) Atka Bay (red marker) is located at the edge of the northeastern Weddell Sea. Coastline data taken from SCAR Antarctic Digital Database. (b) Close-up of map (a) to focus on the study site of Atka Bay. The sampling sites of the standard transect (ATKA) are marked with red circles. (c) Enlargement of (b) showing

in addition to the standard transect (red circles) the parallel transect in the south (ATKA p; light red circles) from ATKA03p
to ATKA21p as well the eastern and western perpendicular transects ATKA S-N (east) from ATKA00SNe to ATKA18SNe
and ATKA S-N (west) from ATKA00SNw to ATKA16SNw, each with a distance of 2 kilometers between adjacent sampling
sites (light blue circles). The southern, eastern and western transects were sampled during a field campaign between November
and December 2018.  The dotted black curves indicate the locations of ice rises and rumples in Atka Bay. Background:
Copernicus Sentinel data 01 December 2018, processed by ESA.
**2.2 Sea-ice conditions**
Atka Bay is seasonally covered with sea ice that is attached to the ice shelf to form immobile fast ice. Following the method
of fast-ice time series retrieval detailed in Fraser et al. (2019), we obtained year-round estimates of fast-ice extent in Atka Bay
from MODIS visible and thermal infrared satellite imagery. Hence, the fast-ice extent time series presented here in Figure 2 is
a) produced at a 1 km spatial and 15 day temporal resolution, from 15-day MODIS cloud-free composite images (following
Fraser et al., 2010) and edge-detected non-cloud-filtered composite images; b) spans the time period from March 2000 to
March 2018; and c) is semi-automated in the sense that the fast-ice edge is automatically delineated during times of high
contrast to offshore pack ice/open water, and manually delineated at other times.
Accordingly, the initial ice formation in the bay has started in March in recent years (Figure 2), with persistent easterly winds
forcing increased dynamic sea-ice growth towards the western ice-shelf edge of the bay. Once the bay is completely covered
by fast ice usually at the end of April (Figure 2), further in-situ ice growth takes place. In the following summer, the ice does
not disappear by surface melting *in-situ*, but breaks up and drifts out of the bay once the conditions are sufficiently unstable.
Stabilization and breakup of the ice-covered bay depend on the presence/absence of pack ice offshore of Atka Bay associated
with changing ocean currents and winds, as well as stationary and passing icebergs. Thus, fast-ice breakup in the bay starts
usually in December/January after the pack ice in front of the fast ice has retreated (Figure 2).
During our study period from 2010/2011 to 2018/19, there were two exceptions to this "typical" annual cycle: In September
2012, a large iceberg (generally referred to as "B15G") grounded in front of Atka Bay, sheltering the fast ice and consequently
preventing sea-ice breakup in the following summer (Hoppmann et al., 2015b), resulting in second-year fast ice in the bay in
2013. A year later, in August 2013, the iceberg dislodged itself, drifting westwards following the Antarctic Coastal Current.
Fragments of the iceberg remained grounded in the northern part of the bay, causing it to be blocked again two years later, and
therefore preventing sea-ice breakup in austral summer 2014/2015 for a second time within the study period. The iceberg
fragments became mobile during the course of the following year, resulting in the bay to become ice-free again in the following
summer.

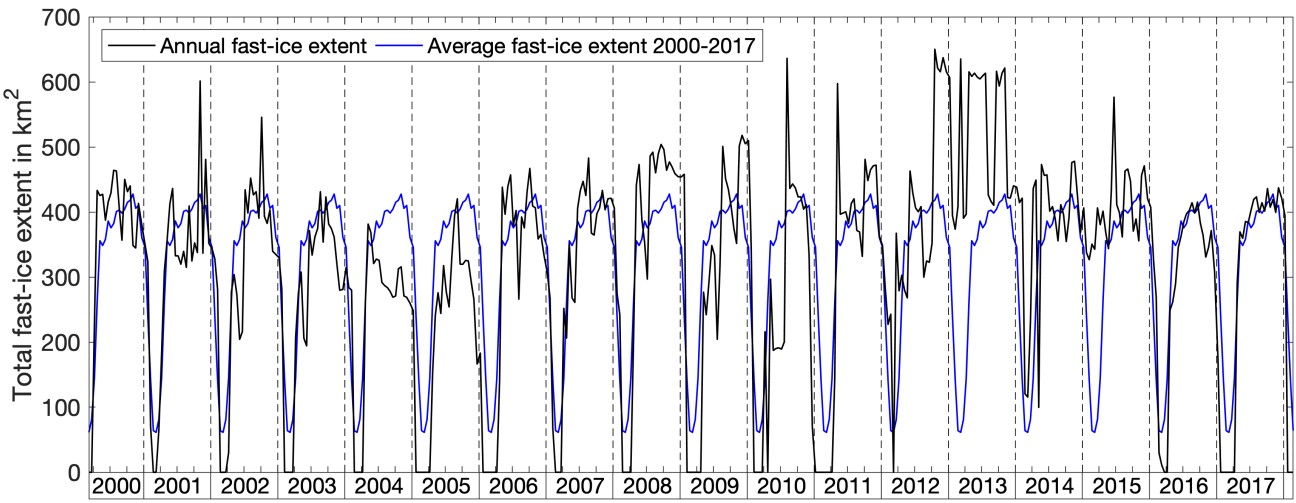

**Figure 2:** Time series of fast-ice extent in Atka Bay between 8° 12'W and 7° 24'W derived from MODIS data between early 2000 to early 2018 (black line). The blue line shows the annual mean extent repeated each year over the same time period. The average fast-ice extent over the entire time series is $319.2 \pm 167.8$ km$^2$, with an uncertainty of 86.6 km$^2$.

## 2.3 Sea-ice measurements across Atka Bay

Since 2010, the AFIN monitoring protocol has been implemented to study the seasonal evolution of fast ice along a 24-km long west-east transect in Atka Bay ("standard transect", red circles in Figure 1). Here, six sampling sites have been regularly revisited between annual sea-ice formation and breakup each year to obtain a continuous record of snow depth, freeboard, sea ice- and sub-ice platelet-layer thickness across the bay (Arndt et al., 2019). Sampling sites on the standard transect are referred to in this paper as ATKAxx, where xx represents the distance in kilometers to the ice-shelf edge in the west.

Generally, measurements along that standard transect are carried out once a month by the wintering team usually between June and January, when safe access to the sea ice is possible. At each sampling site, up to five measurements are taken in an undisturbed area, one as the center measurement and four more at a distance of approx. 5 meters in each direction (north, east, south, and west), in order to account for the spatial variability of sea-ice and snow properties. In years of prevailing second-year ice in the bay (2012/2013, 2014/2015), the number of observations per sampling site was reduced to one (the center measurement) due to exceptionally thick snow and ice. Throughout this manuscript, we mainly present the mean values from those up to five single measurements per sampling site. While all measurements along the standard transect from 2010/2011 to 2018/2019 are included in this study, the sea-ice monitoring activities will be continued beyond this work.

In November and December 2018, additional measurements in both, parallel and perpendicular transect lines to the standard transect, have been performed (Figure 1c). Sampling sites on parallel transects are referred to in this paper as ATKAxxp, where xx represents the distance in kilometers to the ice shelf edge in the west and "p" refers to "parallel". Along the perpendicular

western (w) and eastern (e) transects from south to north, sampling sites are referred to in this paper as ATKAyySNw and
ATKAyySNe, where yy represents the distance in kilometers to the ice-shelf edge in the south.
Sea-ice and platelet-layer thickness as well as freeboard are measured with a (modified) thickness tape. In order to enable the
penetration of the usually semi-consolidated platelet layer, the regular metal plates at the bottom of the thickness tape were
replaced by a metal bar of ~2kg. The underside of the platelet layer is determined by gently pulling up the tape and attempting
to feel the first resistance to the pulling. Sea-ice thickness was measured either by pulling this modified tape through the entire
platelet layer until the solid sea-ice bottom is reached (with a high risk of it getting stuck), or using a regular ice thickness tape.
The modified tape is retrieved by pulling a small rope attached to one side of the metal bar. Snow depth was measured using
ruler sticks. Freeboard is defined as the distance between the snow/ice interface and the sea-water level, while the snow/ice
interface above (below) sea-water level is referred to as positive (negative) freeboard.
In order to determine the influence of snow and the underlying platelet layer to the observed freeboard (F), we also calculated
the freeboard, assuming a hydrostatic equilibrium for floating snow-covered sea ice with an additional buoyancy (the platelet
layer below), using Archimedes' principle:
$$F = -\frac{I \cdot (\rho_I - \rho_W) + S \cdot \rho_S + P \cdot (\rho_P - \rho_W)}{\rho_W} \,.$$  (Eq. 1.1)
As soon as F becomes negative, the involved components of the above-mentioned hydrostatic equilibrium are assumed to be
balanced after the flooding of the snow/ice interface. Here, the depth of the wet soaked snow is considered as equal to the
absolute value of the freeboard. Thus, the latter is calculated for the flooded case as         .
$$F = -\frac{I \cdot (\rho_I - \rho_W) + S \cdot \rho_S + P \cdot (\rho_P - \rho_W)}{\rho_W - \rho_{swet}} \,.$$  (Eq. 1.2)
In Equation 1.1 and 1.2, I refers to sea-ice thickness, S to the dry snow depth, P to platelet-layer thickness, the indices I refers
to sea ice, S to dry snow, P to the platelet layer, and W to water. Constant typical densities of $\rho_W = 1032.3$ kg m$^{-3}$, $\rho_S = 330$
kg m$^{-3}$ and $\rho_I = 925$ kg m$^{-3}$ are assumed in this study. The density of the soaked snow, which is a mixture of water, snow, ice
and air bubbles, is here assumed as $\rho_{swet} = 920$ kg m$^{-3}$ (e.g. Wang et al., 2015). The platelet-layer density $\rho_P$ is calculated by
means of the platelet-layer ice volume fraction $\beta$ as $\rho_P = \beta \cdot \rho_I + (1 - \beta) \cdot \rho_W$. In this study, we used a constant ice-volume
fraction of $\beta = 0.25$, as suggested by Hoppmann et al. (2015b).
The described bore-hole measurements are occasionally complemented by additional total (sea-ice plus snow) thickness
measurements with a ground-based electromagnetic induction instrument (e.g. Hunkeler et al., 2016) as well as autonomous
ice-based systems, such as Ice Mass balance or Snow Buoys (Grosfeld et al., 2015; Hoppmann et al., 2015a). However, this
paper focusses on the regular bore-hole measurements only, as the additional observations address scientific questions beyond
the scope of this paper.

## 2.4 Meteorological conditions and observations at Neumayer III

At the meteorological observatory of the nearby wintering base Neumayer III, atmospheric conditions have been recorded since 1981 (König-Langlo and Loose, 2007), including the study period from 2010/2011 to 2018/19, and continuing beyond it (Schmithüsen et al., 2019). Occasionally, automatic weather stations (AWS) were temporarily installed on the sea ice to record the meteorological conditions directly on the sea ice (Hoppmann et al., 2015a). Since the 2m air temperature and the wind velocity at the meteorological observatory and the AWS on the ice showed a fairly good agreement in prior studies (Hoppmann et al., 2015a; Hoppmann et al., 2013), we use in this paper the more continuous records of the meteorological observatory in order to investigate the links between sea-ice conditions and atmospheric conditions. The Neumayer III data is recorded as minutely averages of typically 10 values per averaging interval. The instrumentation is checked on a daily basis, any erroneous values, e.g. caused by riming or instrument failure, are removed from the record. Therefore, the data quality can be considered high, even though there might be gaps in the records due to the validation routines. Nevertheless, data availability is 99.4% for wind direction, 99.0% for wind speed and 99.7% for air temperature. Uncertainties are essentially those classified by the manufacturers. Instrument details are given in the metadata of the datasets since February 2017 in Schmithüsen et al. (2019), earlier data is documented in König-Langlo and Loose (2007).

Generally, in the vicinity of Neumayer III the weather is strongly influenced by cyclonic activities which are dominated by easterly moving cyclones north of the station. This leads to prevailing persistent and strong easterly winds which exhibit a seasonal cycle with strongest winds during winter time (Figure 3). The second strongest mode in the wind direction distribution at 270° (westward) is associated with super geostrophic flows resulting from a high-pressure ridge north of Neumayer III (König-Langlo and Loose, 2007). These strong winds lead to frequent drifting and blowing snow. Here, we expect snow transport for 10-m wind velocities exceeding 7.7 m/s for dry snow and exceeding 9.9 m/s for wet snow (Li and Pomeroy, 1997).

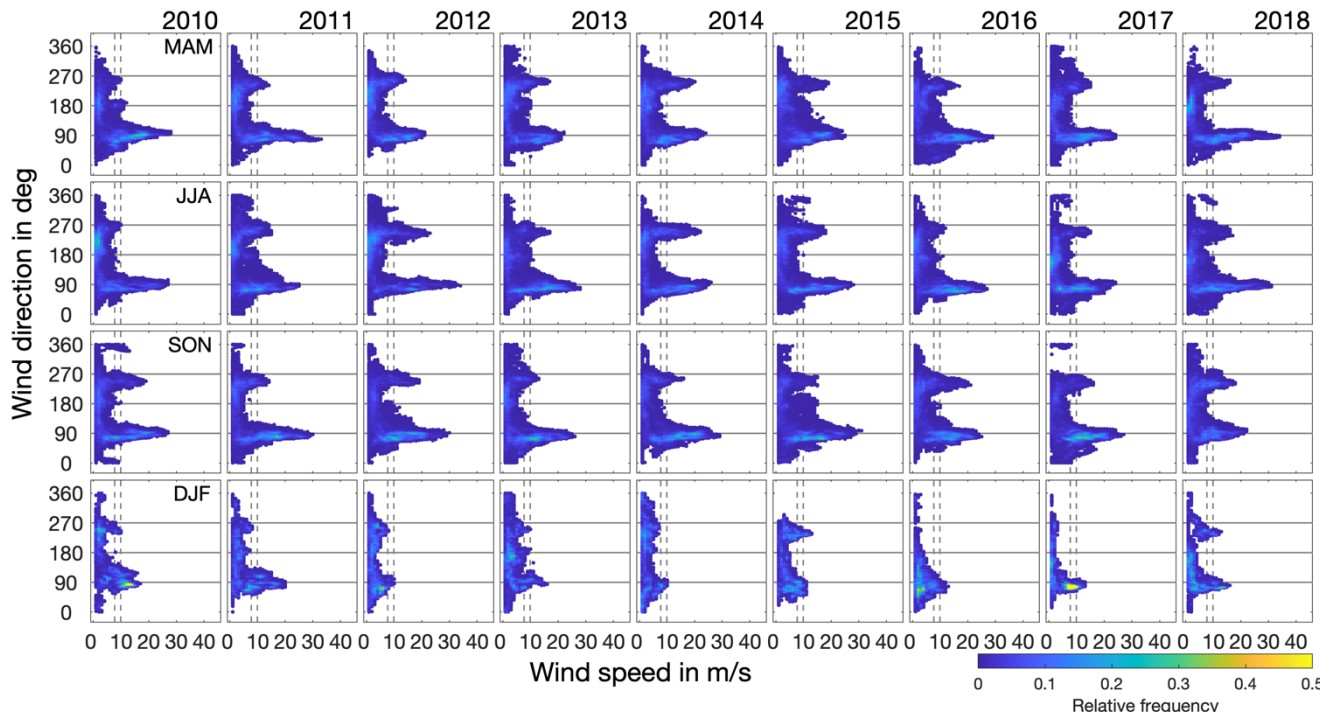

**Figure 3:** Distribution of wind speed related to wind directions separated for austral fall (March, April, May; MAM), winter (June, July, August; JJA), spring (September, October, November; SON), and summer (December, January, February; DJF) for the study period from 2010 to 2018. Colors indicate the relative frequency of each shown wind direction to wind speed pair. Dashed vertical lines denote thresholds for 10-m wind speeds for snow transport of dry (7.7 m/s) and wet snow (9.9 m/s) (Li and Pomeroy, 1997).

## 3 Results

### 3.1 Nine-year record of sea-ice and platelet-layer thickness, snow depth and freeboard along a 24-km W-E transect

Figure 4 summarizes all conducted measurements of snow depth, sea-ice and platelet-layer thickness on the standard transect from bore-hole measurements for each ATKA sampling site in the study period from 2010 to 2018. In the seven months when sea-ice conditions allowed safe access (usually from May/June to December), about eight sets of measurements were taken along the standard transect crossing Atka Bay, i.e. once every three to four weeks.

Analyzing the average annual maximum values of the investigated parameters (Table 1) for years of seasonal fast ice only (excluding 2013 and 2015) and neglecting local iceberg disturbances (ATKA07 in 2017), the highest annual snow accumulation of $0.89 \pm 0.36$ m was measured at ATKA07, while the smallest by far was measured at ATKA24 at the easternmost sampling site, with only $0.28 \pm 0.19$ m. Averaged over the entire bay, the lowest snow accumulation of $0.51 \pm 0.30$ m was observed in 2016. In contrast, 2011 was the year with the most snow and an average snow depth of $0.85 \pm 0.20$ m

across the bay. The average seasonal fast-ice thickness based on the measurements during the observation period varied between $1.74 \pm 0.31$ m (ATKA21) and $2.58 \pm 1.28$ m (ATKA07) with a mean value of $1.99 \pm 0.63$ m. The underlying seasonal platelet layer reached an average annual thickness of 3.91 m, which, however, shows a strong gradient in the average annual maximum values (Table 1) from $4.62 \pm 0.67$ m at ATKA07 in the west of the bay to $2.82 \pm 1.20$ m at ATKA24 in the east. In 2013 and 2015, the fast ice in Atka -Bay became second-year ice due to grounded icebergs in front of the bay. Within the respective second year, snow depth increased further by an additional $0.88 \pm 0.43$ in 2013 and by $0.74 \pm 0.27$ m in 2015. In 2013, the average fast-ice thickness across the bay increased by an additional $1.21 \pm 0.42$ m, while in 2015, it increased by an additional 2.79 m $\pm 1.48$ m. In the years of prevalent second-year ice in the bay, the thickness of the platelet layer increased on average by 5.13 m $\pm 1.43$ m in 2013 (compared to the end of 2012), and 4.11 m $\pm 1.86$ m in 2015 (compared to the end of 2014). During these periods, ATKA11 experienced the highest annual platelet-layer thickness increase of 6.82 m and 6.44 m, respectively.

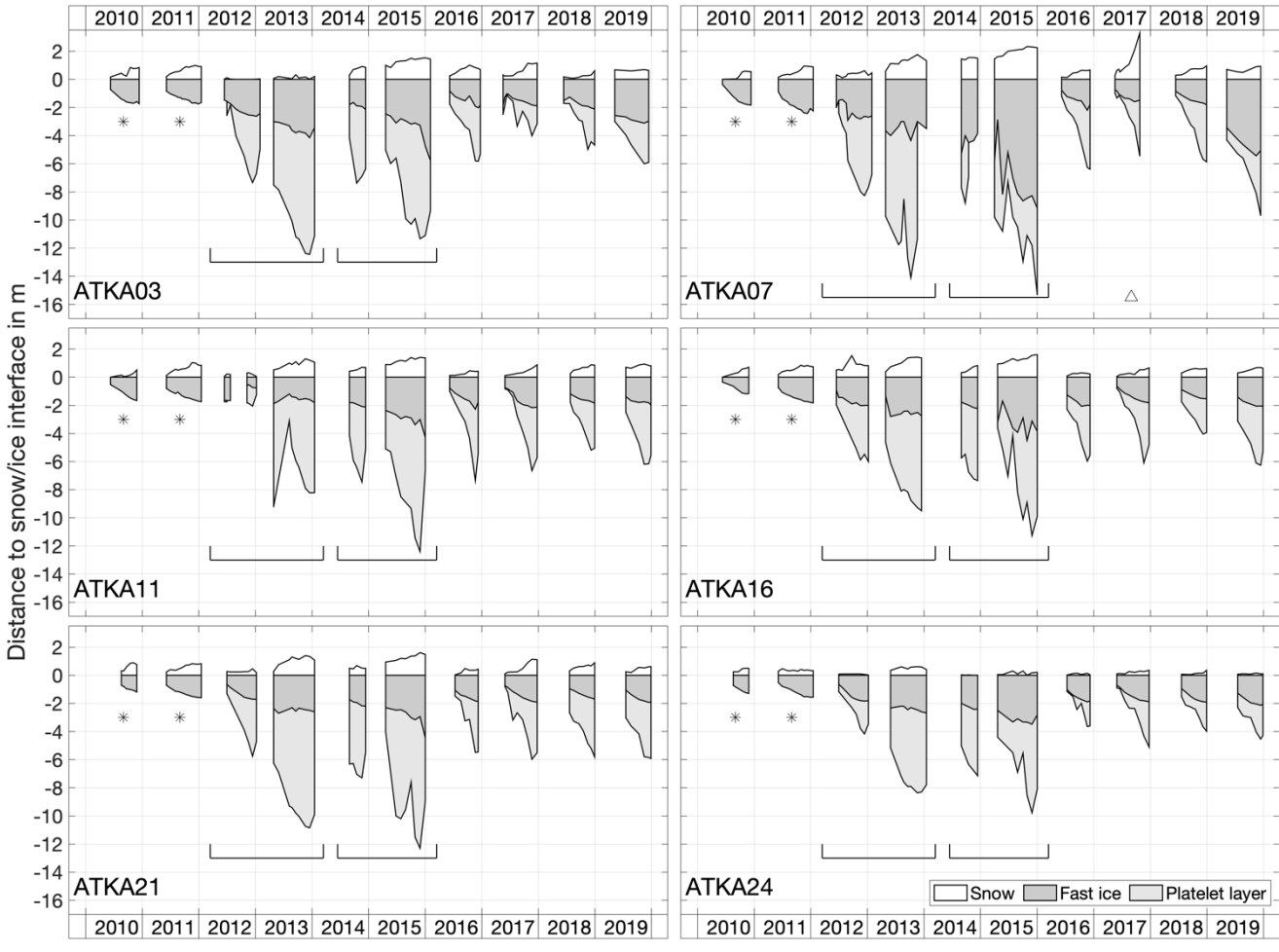

**Figure 4**: Time series of snow depth, fast-ice and platelet-layer thickness from bore-hole measurements along the standard transect for each ATKA sampling site (Figure 1) for the time period from 2010 to 2018. Note: In 2010 and 2011, the platelet-layer thickness was not measured (\*). In 2012/2013 and 2014/2015 Atka Bay was blocked by icebergs, so the fast ice did not break up and turned into second-year ice instead (↵). In 2017, a small iceberg in the vicinity of ATKA07 strongly influenced the snow measurements (△). Reference depth of 0 meters is the snow/ice interface.

Figure 5 depicts the evolution of the water level with respect to the snow/ice interface (which is the freeboard with an opposite sign) along the standard transect in the study period from 2010 to 2018. Taking all conducted freeboard measurements from seasonal fast ice into account, 38% reveal negative data, i.e. flooding can be assumed, with an average negative freeboard of -0.10 ± 0.08 m. In contrast, considering freeboard measurements from second-year ice only, 55% of the data indicate a negative freeboard, with an average of -0.22 ± 0.15 m. Analyzing the average annual maximum of the negative freeboard values (Table 1) for years of seasonal fast ice only, and neglecting local iceberg disturbances (ATKA07 in 2017), there is no distinct gradient across Atka Bay, but higher average negative freeboard values (-0.07 to -0.08 m) are recorded both in the far west (ATKA03) and in the east (ATKA16 and ATKA21), whereas the lowest average negative freeboard of -0.01 ± 0.08 m was measured at ATKA07. According to Equation 1.1, 70% of the calculated freeboard values are smaller than the measured values. The difference between measured and calculated freeboard values ranges from -0.54 to 1.26 m with an average of -0.02 ± 0.18 m. Neglecting the underlying buoyant platelet layer in the calculation reduces the freeboard by 0.03 ± 0.17 m, whereas neglecting the snow layer on top of the sea ice increases the freeboard by 0.17 ± 0.25 m (Figure 5).

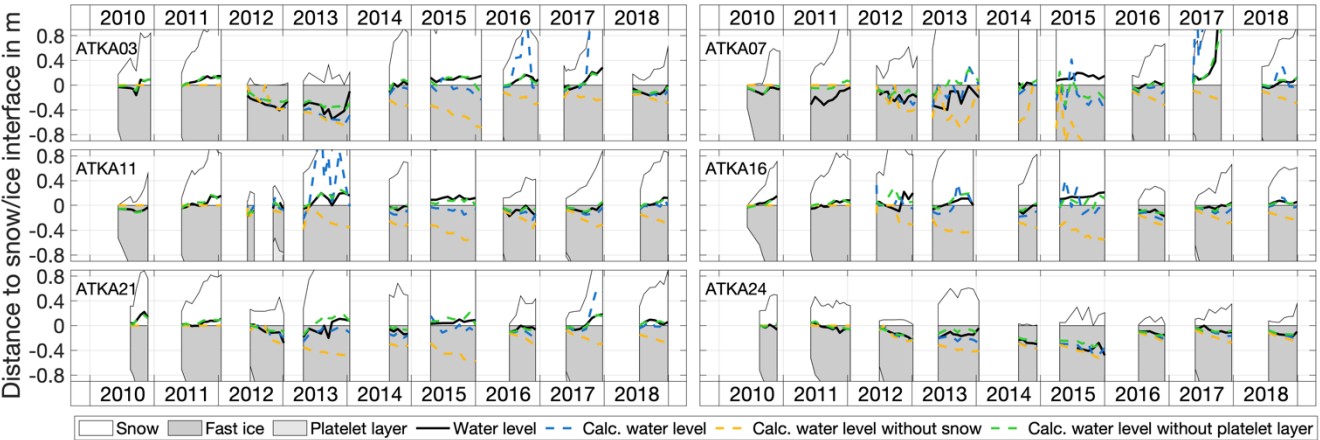

**Figure 5:** Close-up of Figure 4 which highlights the location of the water level with respect to the snow/ice interface (which has the opposite sign of the freeboard) as measured in the field (black solid line) and as calculated according to Equation 1.1 and 1.2 including snow and platelet-layer thickness (blue dashed line), neglecting the snow cover (dashed yellow line) and platelet layer (dashed green line), respectively. Please note that, for the purpose of better illustration, we depict here the actual

location of the water level rather than the freeboard (the only difference being the opposite sign). This means that, if the water level is above the snow/ice interface, this is depicted in the figure accordingly, while the actual freeboard carries a negative sign, and vice versa. The reference depth of 0 represents the snow/ice interface.

**Table 1:** Average annual maximum of snow depth, sea-ice and platelet-layer thickness, as well as freeboard (negative equals potential flooding) on the standard transect from bore-hole measurements for each ATKA sampling site (Figure 1) for the time period from 2010 to 2018, excluding years of second-year ice due to blocking of the bay (i.e. 2013 and 2015). [1]At ATKA11 all measurements of the year 2012 are also neglected as the ice has temporarily broken up again. [2]At ATKA07 the snow measurements of the year 2017 are also neglected as a small iceberg has strongly influenced the accumulation rates. Standard deviations are given in parentheses.

| | ATKA03 | ATKA07 | ATKA11 | ATKA16 | ATKA21 | ATKA24 |
|---|---|---|---|---|---|---|
| Snow depth in m | 0.81 (0.35) | 0.89 (0.36)[2] | 0.74 (0.23)[1] | 0.79 (0.37) | 0.77 (0.24) | 0.28 (0.19) |
| Ice thickness in m | 2.04 (0.31) | 2.58 (1.28) | 1.97 (0.25)[1] | 1.81 (0.36) | 1.74 (0.31) | 1.83 (0.35) |
| Platelet-layer thickness in m | 3.88 (1.31) | 4.62 (0.47) | 4.59 (0.83)[1] | 3.99 (0.94) | 4.21 (0.54) | 2.82 (1.20) |
| Freeboard in m | -0.08 (0.14) | -0.01(0.08)[2] | -0.05(0.08)[1] | -0.07 (0.09) | -0.08 (0.10) | -0.05 (0.09) |

**3.2 Seasonal snow, sea-ice and platelet-layer accumulation/growth and melt rates**

Figure 6 summarizes the seasonal snow depth, sea-ice and platelet-layer thickness evolution separated for austral fall (March, April, May; MAM), winter (June, July, August; JJA), spring (September, October, November; SON), and summer (December, January, February; DJF) averaged for each ATKA sampling point over the duration of the whole study period from 2010 to 2018.

Considering the average monthly snow accumulation rates, a slight increase from fall (from 0.04 to 0.09 m per month, across stations) to spring (0.09 to 0.11 m per month) becomes apparent, if excluding the eastern sampling sites at ATKA21 and ATKA24. Latter sampling sites show the highest monthly averaged accumulation rates during austral fall (0.11 and 0.08 m per month), which subsequently decrease to 0.10 and 0.02 m per month, respectively. In contrast, a clear snow loss with a maximum monthly average of up to $0.03 \pm 0.12$ m at ATKA11 and a maximum snow loss rate of 0.21 m per month at ATKA07 (80[th] percentile), can be seen mostly during summer months. The seasonal evolution of the platelet layer shows a similar pattern: between austral autumn and spring, an average monthly thickness increase of up to $0.82 \pm 0.30$ m at ATKA11 is

observed. Excluding ATKA07, afterwards an average monthly platelet-layer thickness decrease of 0.85 m is calculated for summer. The maximum decrease of 6.25 m per month occurred at ATKA11 in 2013 (80th percentile). However, it is highly likely that this is a measurement error. In contrast, ATKA07 also reveals an increase in platelet-layer thickness during the summer months with a monthly average of $0.67 \pm 1.20$ m. With regard to the growth rates of fast ice in Atka Bay, a contrasting but expected seasonal development is observed: The highest average monthly fast-ice growth rates of up to approx. 1 m per month (80th percentile) are measured in autumn, and decrease in the following month until spring. These exceptionally high growth rates result from rapid growth of the solid fast ice into the (unconsolidated) sub-ice platelet layer, i.e. from the subsequent freezing of the interstitial water between the platelets in the top part of the platelet layer. In other words, some of the heat within the newly growing ice was already extracted earlier by the ocean during the process of platelet crystal formation in the supercooled Ice Shelf Water plume. In the subsequent summer months, average monthly sea-ice growth rates increased again to values between 0.07 m (ATKA07) and 0.24 m (ATKA21), except for ATKA24, where sea-ice melt dominates with an average monthly melt rate of $-0.05 \pm 0.22$ m and a maximum monthly sea-ice melt rate of -0.58 m.

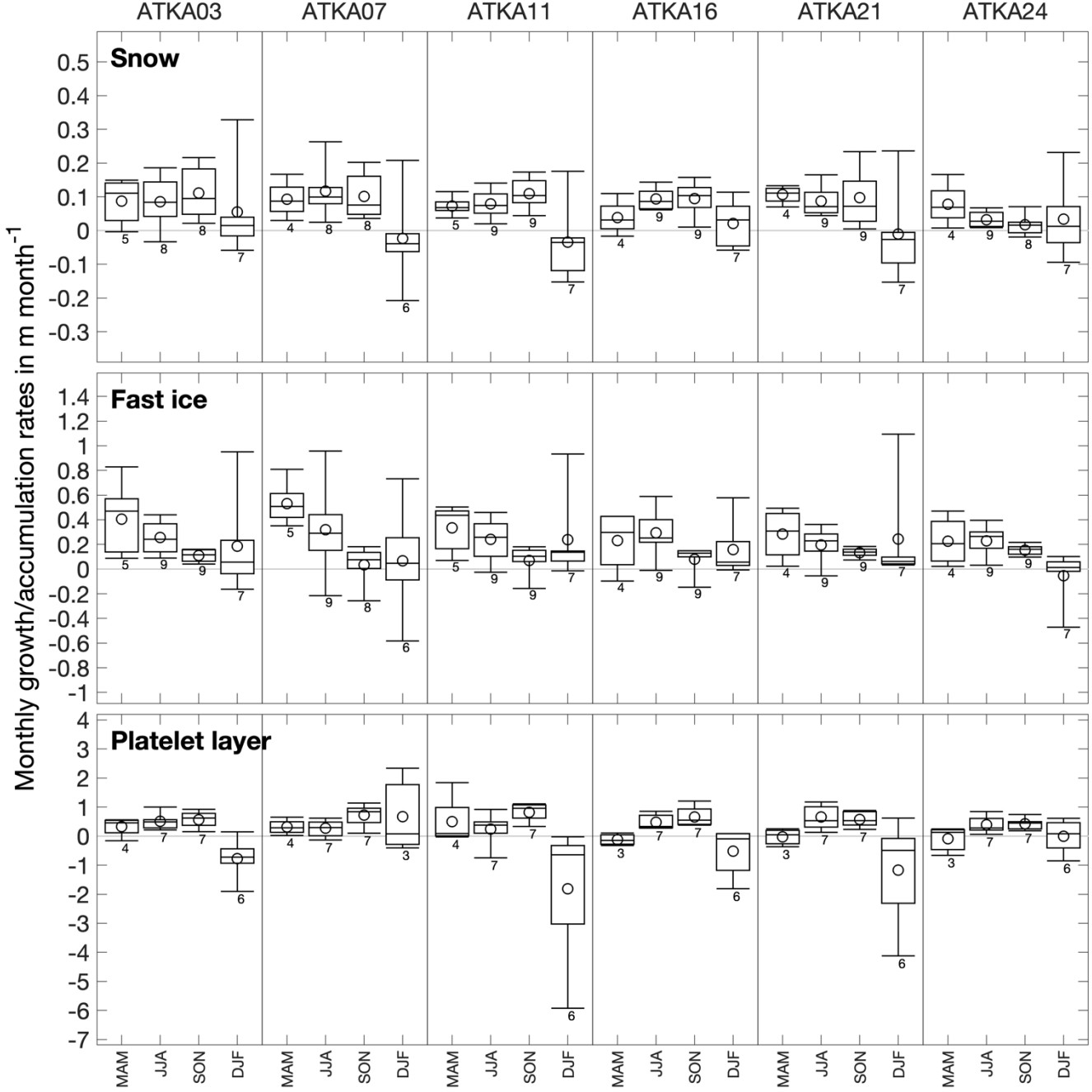

**Figure 6:** Seasonal snow, sea-ice and platelet-layer accumulation/growth and melt rates separated for austral fall (March, April, May; MAM), winter (June, July, August; JJA), spring (September, October, November; SON), and summer (December, January, February; DJF) for each ATKA sampling point for the study period from 2010 to 2018. Boxes are the first and third

quartiles; whiskers the 20th and 80th percentile. Circles indicate the mean, horizontal lines in the boxes the median. Numbers
below the whiskers indicate the respective sampling size, i.e. the number of included years, with a maximum of nine.

### 3.3 Spatial variability of snow depth, sea-ice and platelet-layer thickness

In order to describe the spatial variability of snow depth, sea-ice and platelet-layer thickness in west-to-east as well as in south-
to-north direction across Atka Bay, additional parallel and perpendicular transects to the standard transect have been sampled
in November/December 2018 (Figure 7). Considering the solid sea ice only, the complementary transect data show that sea-
ice thickness over the bay in south-north and west-east direction is rather constant with an average of $1.68 \pm 0.21$ m. In contrast,
neglecting the measurements in iceberg-affected areas, snow depth data show higher values in the south and in the center of
the bay of up to $1.00 \pm 0.04$ m, while decreasing significantly towards the towards the ice-shelf edges in the east and west and
northern fast-ice edge to $0.08 \pm 0.01$ m and $0.28 \pm 0.09$ m, respectively. The platelet-layer thickness beneath the fast ice shows
a large spatial variability. While all measurements on the standard transect reveal the lowest platelet-layer thickness in the east
of the bay at ATKA24 (see Section 3.1), on the parallel transect in the south a maximum platelet-layer thickness of $7.18 \pm 0.26$
m at the easternmost sampling point (ATKA21p) is observed. For the perpendicular transects in south-to-north direction, a
significantly decreasing gradient in platelet-layer thickness from the ice-shelf edge towards the northern fast-ice edge is
evident. On the western south-to-north transect, a decrease from $6.62 \pm 0.25$ m to $2.33 \pm 0.08$ m was observed, whereas for
the eastern transect this strong gradient is even more apparent with a decrease from $9.17 \pm 0.11$ m to $1.88 \pm 0.20$ m.

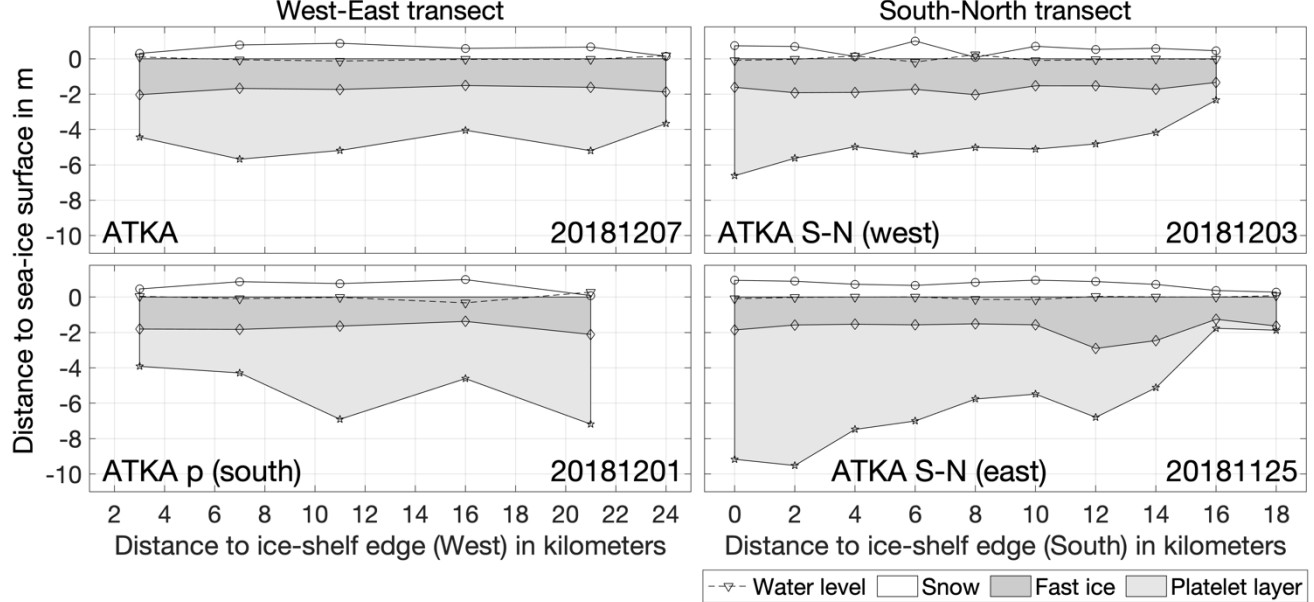


**Figure 7:** Overview of measurements on the standard transect from west to east (upper left), the parallel one (lower left), the western perpendicular transect from south to north (upper right) and the respective parallel one to the east (lower right) showing the water level, snow depth, fast-ice and platelet-layer thickness across Atka Bay. All measurements were conducted between November 25, 2018 and December 07, 2018. For the parallel west-east transect (December 01, 2018), the platelet-layer thickness evolution is influenced by a nearby iceberg (see Figure 1c). Also, for the western north-south transect (December 03, 2018), snow measurements are influenced by several small icebergs in the vicinity between kilometers 4 and 8 (see Figure 1 c).

## 4 Discussion

### 4.1 Seasonal and interannual variability of snow depth, sea-ice and platelet-layer thickness

The fast-ice regime in Atka Bay is primarily seasonal and the sea-ice cover usually only remains in the bay if a breakup is prevented by grounded icebergs in front of it. For example, while in 2013 a 17km-by-10km iceberg (B15G) blocked the entire bay, in 2015, only small iceberg fragments of B15G in front of the bay were sufficient to ensure that the sea ice in the bay did not break up, but became perennial. It may also occasionally happen that small areas of fast ice remain attached to the ice-shelf edge, or that individual ice floes remain in the bay and are incorporated into the newly growing ice in the following winter. Not only does the presence and size of the icebergs play a role in the fast-ice seasonality, but also the location and associated influence of atmospheric circulation patterns and ocean processes.

Considering first of all the seasonal sea ice only, the presented measurements along the standard transect across Atka Bay indicate a clear seasonal cycle in all investigated variables, i.e. snow depth, sea-ice and platelet-layer thickness: The initial sea-ice formation in Atka Bay starts in March and proceeds towards a completely fast-ice-covered bay at the end of April. The continuous sea-ice growth (i.e. ocean-atmosphere heat flux) proceeds with decreasing growth rate through fall and winter until the thickening snow cover more and more reduces the heat flux between the upper ocean and the atmosphere, preventing further thermodynamic sea-ice growth. However, the fast-ice thickness still increases in spring and even during austral summer months (albeit very slowly). This can be explained by the measurement uncertainty with respect to the large spatial variability of sea-ice thickness even on very small (centimeter) scales, but the consistency in the data suggests that it could also be caused by consolidation processes within the platelet layer below, i.e. in-situ sea-ice growth by heat transport into a supercooled plume residing right beneath the solid fast ice similar to observations in McMurdo Sound (Smith et al., 2012; Leonard et al., 2011; Dempsey et al., 2010; Robinson et al., 2014). So far, in Atka Bay there is only evidence that platelets grow quite large already while still suspended in the water column (Hoppmann et al., 2015b). To what degree an in-situ growth of platelet crystals and consolidation processes that go beyond regular freeze-in of the topmost part of the platelet layer by heat conduction to the atmosphere play a role at Atka Bay still needs to be investigated. In any case the platelet layer is an efficient buffer between the fast ice and the incoming warmer water in summer (Eicken and Lange, 1989), so the lack of noticeable fast-ice bottom melt is generally expected. Oceanographic (winter) data is sparse, and the monitoring at Atka Bay has recently been

extended to also include regular CTD casts, whenever the (challenging) conditions and time constraints allow. An analysis of available CTD data in Atka Bay is currently ongoing, and will be shown in a future dedicated study to close the above observational gaps with respect to the ocean.

Destabilization of the fast ice and the platelet layer below in summer is to a large extent driven by the presence/absence of pack ice offshore Atka Bay. Thus, the initial breakup and subsequent retreat of the pack ice in front of the bay allow for locally increasing ocean currents beneath the fast ice and the inflow of warm Antarctic Surface Water from the east (Hoppmann et al., 2015a; Hattermann et al., 2012) causing both washing out of the platelet layer as well as mixing warm water into the water column associated with a thinning rate of the platelet layer of approx. one meter per month from December onwards. The retreating fast ice also initializes the sea-ice breakup in the bay starting usually in December/January (Figure 2). The diminishing fast-ice zone potentially goes along with an additional thinning of the platelet layer (Figure 8 b) by, e.g., further washing out mechanisms. Even though the correlation between the change of fast-ice extent and the mean (maximum) platelet-layer thickness between two consecutive surveys with an r-coefficient of 0.38 (0.35) is relatively low, Figure 8b indeed suggests that decreasing platelet-layer thicknesses are generally associated with retreating fast ice in Atka Bay. Also, Massom et al. (2018) have also shown that pack ice has a stabilizing effect as a buffer against ocean swells. The deployment of complex oceanographic moorings, either ice- or seafloor-based, would greatly help to further investigate ocean properties and currents and their effect on sea ice and the ice shelf, but their deployment, and especially their recovery, is extremely difficult and risky in the dynamic and harsh conditions of Atka Bay. While such deployments are logistically not feasible at the moment, it is planned to include suitable instrumentation in the monitoring within the next years.

In contrast to the decrease in platelet-layer thickness beneath the fast ice in summer over the entire bay, the snow cover on top does not show a clear seasonal pattern, but indicates a decrease in snow depth with increasing air temperature. However, even in summer, no consistent snow melt with associated strong mass loss is observed over the entire bay. Rather, a strong variability in snow depth over all sampling sites and all sampling years with a weak snow loss during summer months is observed (Figure 6). The latter pattern is a result of both, temporary temperatures above freezing which favor surface melting (Figure 8a), and comparatively low wind speeds (Figure 3) preventing the accumulation of additional snow blown over from the surrounding ice shelf. These results match well with results from studies on the seasonal cycle of snow properties in the inner pack ice zone of the Weddell Sea as for example performed by Arndt et al. (2016), who also showed missing persistent summer melt as highlighted above.

Overall, the 9-year time series for in-situ snow depth, sea-ice and platelet-layer thickness in Atka Bay do not show any trend over the analyzed study period, whereas their inter-annual variability is dominated by local or temporary effects such as the presence of icebergs, which may for example lead to small-scale strong snow accumulations (Figure 4) or occasionally even to a perennial fast-ice regime. It is particularly remarkable that the average annual maximum platelet-layer thickness of 4 m (Table 1) is consistent with an earlier investigation at Atka Bay performed in 1982 by Kipfstuhl (1991) between the western ice-shelf edge and ATKA03, and, at the same time, much higher than in results from another study in 1995, where a maximum platelet-layer thickness of 1.5 m was measured in a similar location (Günther and Dieckmann, 1999). Considering a) the fact

that these two studies only sampled one location, and b) the generally large spatial and temporal variabilities of the platelet-
layer thickness, and in the present more detailed study, we infer that there seems to be no clear trend over the past decades.
Our results are also in line with a recent study of Brett et al. (2020), who also found spatially highly variable platelet layers of
4+m under fast ice in McMurdo Sound. Thereby, our results suggest that it is likely that the relevant (sub ice-shelf) processes
in this region have not changed much either. However, as already stated above, it is crucial to further look into all the available
oceanographic data that are available from the Atka Bay region in order to support (disprove) these indications provided by
the fast-ice monitoring. Another benefit of such comparison would be to strengthen (weaken) the hypothesis that fast-ice
properties can serve as an indicator for the status of an ice-shelf, as suggested by Langhorne et al. (2015).

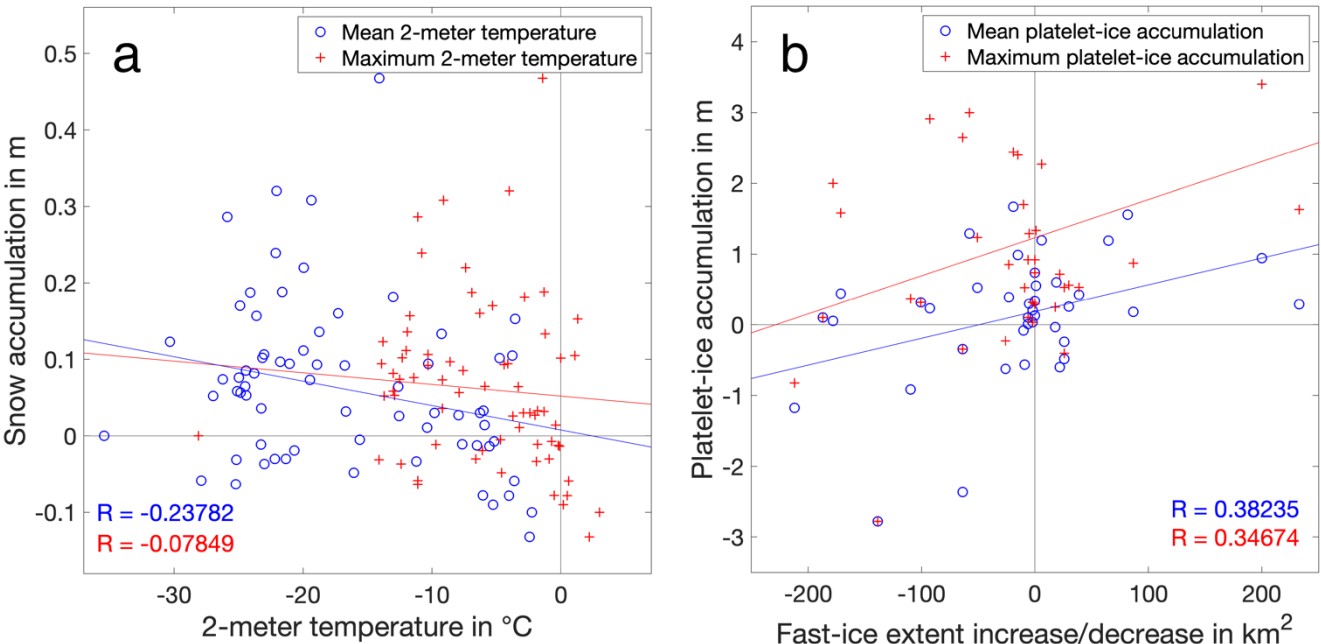

**Figure 8:** Scatter plot comparing (a) the average 2-meter air temperature (see Section 2.4) and the snow accumulation between
two consecutive surveys, and (b) increasing (positive values) and decreasing (negative values) fast-ice extent and platelet-
layer thickness between two consecutive surveys. The analysis includes all measurements at all sampling sites throughout the
study period from 2010 to 2018. Blue circles and red crossed denote the respective mean and maximum values within the time
frame between the consecutive measurements. Colored solid lines in Figure (b) show the linear regression between both
parameters with the respective correlation coefficients R.
**4.2 Spatial variability of fast-ice properties related to the distance to the ice-shelf edges around the bay**
When neglecting local disturbances, such as icebergs, our results clearly indicate differences in the evolution of platelet-layer
thickness and snow depth with respect to the distance to the adjacent ice-shelf edges around Atka Bay. In contrast, the fast ice

itself does not exhibit any large spatial variability, with at the end of the season a nearly uniform thickness of 2 meters across the bay, both in west-east and south-north direction (Figure 7).

Analysis of the spatial distribution of platelet-layer thickness under the fast ice along the standard transect over the entire bay reveals that ATKA24 shows a significantly thinner platelet layer than all other sampling sites. In contrast, the parallel transect towards the south reveals a significantly higher platelet-layer thickness at the sampling point closest to the eastern shelf ice edge (ATKA21p). Perpendicular sampling transects from close to the southern ice-shelf edge towards the fast-ice edge in the north show a strong increase of platelet-layer thickness near the ice-shelf edge, followed by a moderate decrease in platelet-layer thickness towards the north, which rapidly decreases about 5 kilometers off the fast-ice edge. This thickness gradient is much more pronounced on the south-north transect in the central area of the bay compared to the western one. Moreover, considering the entire time series of the west-east transect (Figure 4), the highest platelet-layer thickness is observed in the central area of the bay (ATKA07 and ATKA11). Summarizing all these observations, we hypothesize that, on the one hand, the strongest outflow of supercooled water from the ice-shelf cavity (along with associated suspended platelet crystals) leads from the south centrally into the bay. On the other hand, local under-water topographic features of the ice shelf (i.e. ice rises) at the eastern boundary of Atka Bay (Figure 1c) might lead to a blocking of ocean currents and thus the advection of suspended platelet crystals, causing the high platelet-layer thickness at ATKA21p and the consistently low observed thickness north of this location at ATKA24 (Figure 4). The strongly decreasing gradient in the platelet-layer thickness towards the northern sea-ice edge is likely due to increasing distance from the source of suspended platelet crystals being advected from under the ice shelf and related washout effects. This is especially likely since at the time of the corresponding measurement, the pack ice in front of the bay was already broken open, allowing for wind-induced currents and locally solar-heated water production (the so-called mode 3 incursions, Jacobs et al. (1992), their Figure 1). Also, the fact that the northernmost sampling points are located close the edge of the bay or even already outside of it, raises the probability that the predominant coastal ocean current transports warm Antarctic Surface Water towards the fast-ice area, which would further intensify this effect (Hattermann et al., 2012; Hoppmann et al., 2015b). From this, it can be generalized that a smaller amount of platelet crystals can accumulate under narrow fast-ice areas, since these are exposed to stronger oceanic currents and associated washout effects, as well as warm water incursions. In that respect, Hoppmann et al. (2015b) used a subset of oceanographic data collected by the nearby PALAOA hydrographic observatory (Boebel et al., 2006) to link fast ice observations to ocean properties. A more recent study by Smith et al. (2020) helped to constrain the boundary conditions for Ice Shelf Water outflow by mapping in great detail the cavity geometry of the Ekström Ice Shelf. This study also shows data from repeated CTD casts through a borehole in the ice shelf, revealing the buoyant outflow of Ice Shelf Water in a relatively shallow surface layer. While these efforts help to better understand the complex system of ice shelf-ocean-sea ice interaction in this region, we suggest that a more comprehensive, year-round oceanographic study also implementing a dedicated survey program is still urgently needed as a complement to the ongoing sea ice monitoring. This would allow us to investigate in more detail the outflow of Ice Shelf Water and the complex processes involved in the redistribution of platelet crystals that emerge from the ice shelf cavity. Comparative

analyses to other study regions are not possible at this time, since, to our knowledge, no comparable transects were carried out so far in other Antarctic fast-ice regions with platelet layers beneath.

Examining the spatial distribution of snow over the bay, the considerably lower snow depth at ATKA24 compared to all other sampling sites is striking (Figure 4), and most likely related to the proximity to the ice-shelf edge in approximately 1 km distance. Due to the prevailing easterly winds in the bay (Figure 3), an east-west gradient in snow depth could have been expected over the rest of the bay. However, this gradient cannot be determined on average over the entire time series. This is mainly due to temporary local disturbance factors in the bay, such as icebergs and pressure ridges, which locally dominate the snow distribution and thus lead to a comparatively homogeneous distribution of snow depth over the central part of Atka Bay. A south-north survey across the bay at the beginning of austral summer 2018, however, revealed a trend of decreasing snow depth towards the northern fast-ice edge, with a stronger gradient approx. 5 km from the ice edge (Figure 7), which is in line with the northern boundary of the ice-shelf edge (Figure 1) and can therefore be explained by associated decreasing effects of the prevailing offshore winds. Consequently, also the measured snow depth is less in that part of the bay.

Due to the generally thick snow cover on Antarctic sea ice (Kern and Ozsoy-Çiçek, 2016; Markus and Cavalieri, 1998; Massom et al., 2001b), flooding of the snow/ice interface and the resulting formation of snow-ice is a widespread phenomenon in the Southern Ocean and contributes significantly to the sea-ice mass budget in the area (Eicken et al., 1995; Jeffries et al., 2001). While Günther and Dieckmann (1999) observed no flooding in Atka Bay during their study, Kipfstuhl (1991) reported flooding in relation to snow loads greater than 1 meter, an observation that we can largely confirm with our data. Exceptions are measurements on comparatively thin ice that already showed a sufficient snow layer, e.g. in the austral winter 2010 and 2011 at ATKA21, leading to a negative freeboard and potential flooding already early in the season (Figure 5). Consequently, taking all conducted freeboard measurements on the seasonal fast ice into account, 55 % of the data indicate a negative freeboard, i.e. potential flooding and associated snow-ice formation can be assumed. While the snow cover reduces the buoyancy of the sea ice and accelerates flooding, the underlying platelet layer counteracts this by adding additional buoyancy. However, neglecting the platelet layer reduces the freeboard by $0.09 \pm 0.06$ m, but still a negative freeboard is derived in half of the calculations. Thus, the spatial distribution of the sign of the freeboard, and therefore also the flooding of the snow/ice interface, is essentially controlled by the snow layer on top of the fast ice in Atka Bay. The thickness of the underlying platelet layer below, in turn, contributes to the resulting thickness of the flooded layer and consequently to the thickness of the potential snow-ice layer.

**4.3 Impact of local disturbances on bay-wide properties and processes**

As already stated above, the largest overall effect on the fast-ice properties, the underlying platelet layer and the snow on top is due to the presence of icebergs in front of Atka Bay, which might entirely prevent a fast-ice breakup. At the same time, those grounded icebergs that are enclosed by sea ice within the bay add strong local effects. Thus, the large iceberg B15G grounded in front of Atka Bay sheltering the fast ice in the bay and consequently preventing sea-ice breakup in the following summer (Hoppmann et al., 2015b) led to second-year fast ice in the bay in 2013. Our measurements have shown that this hardly had any effect on the monthly snow accumulation rate and platelet-layer growth rate, but rather that these were within

the same range as in the years of seasonal sea-ice cover. Accordingly, for second-year sea ice, the total annual snow and
platelet-layer thicknesses are approximately twice as thick as in the other years and average to $1.30 \pm 0.60$ m and $7.84 \pm 1.33$
m, respectively. Higher snow loads do also increase the probability and extent of surface flooding. This is not only observed
for years of second-year sea ice, but also for local disturbances as a result of the presence of small icebergs inside of the bay.
In contrast, the evolution of the second-year fast-ice thickness shows a different pattern: Considering the large sea-ice thickness
of around 2 m, as well as the insulating effect of the thick snow cover on top, the contribution of congelation growth is very
limited. Instead, it is highly likely that dynamical growth as well as growth related to the consolidation of the platelet layer
dominates the thickening of the perennial fast ice, adding up to an average thickness of $4.19 \pm 1.90$ m, which is even more
than double the thickness of seasonal sea ice in the bay.

## 4.4 Sea-ice growth history

A detailed study of sea ice crystal fabric by means of visual inspection of thick/thin sections or with the help of an automated
fabric analyzer can help greatly to determine the dominant growth processes in a given area of interest. At the same time, the
growth history of fast ice is to a large degree governed by the timing of the formation of a persistent ice cover, and can only
be interpreted accurately by the help of as much auxiliary information as possible, most importantly from regular satellite
imagery such as MODIS, Sentinel-1 or Radarsat.
It has been planned since the start of the AFIN monitoring at Atka Bay in 2010 to regularly obtain sea ice cores for crystal
fabric analysis. A set of cores from the six main sampling sites (Figure 1) has been obtained in 2011, and again in 2012. Only
4 out of these 12 cores have been processed so far (all from 2012), which is obviously only a very small sample size compared
to the decade of measurements shown above. While the limited ice core data thereby is insufficient to make general statements
about sea ice growth processes at Atka Bay, we provide this data here to highlight a few major aspects, some of which have
already been discussed earlier.
From the (limited) data we have from the four 2012 cores (Figure 9), it is evident that 1. there is no columnar texture at all; 2.
there is a small fraction of granular ice in the top parts of three cores; 3. there is a small fraction of snow ice in one core and
4. all cores are dominated by incorporated platelet ice. The core from the western part of Atka Bay (ATKA03) exhibits a
comparably high fraction of granular ice: a 0.5m long section at the top, and 2 smaller sections a little bit deeper, with some
incorporated platelet ice in between. This crystal fabric is a manifestation of the dynamic conditions under which the initial
growth takes place, and supports the other datasets shown above. The strong easterly winds (Figure 3) keep pushing the initially
forming thin ice towards the western ice shelf edge, which leads to a grinding of the fragile frazil crystals, and subsequently
to a rafting of the newly formed ice. This process seems to be still relevant even after the ice has thickened to >0.5 m, probably
by very strong winds. In this way, the thickening rate of the sea ice is greatly accelerated initially (Figure 4). The absence of
exclusively columnar ice is evidence that there are already platelet crystals emerging from the cavity very early in the season.
While it has been suggested in an earlier study that such crystals would be present in the bay from June onwards (Hoppmann
et al., 2015b), there is a possibility that they might arrive even earlier, at least in parts of the bay close to the outflow of ISW.
While the ice core taken at ATKA11 is not representative at all for sea ice in the bay due to an early breakup event and
subsequent late refreezing, the presence of snow ice is an evidence for a process that we would argue plays an underestimated
role in this region. However, we currently do not have any more direct evidence for the wide presence of snow ice at Atka Bay
(due to the lack of ice core data) other than the observations of negative freeboard in our main dataset (Figure 5), and several
observations of extensive surface flooding from summer campaigns (although the latter do not necessarily result in snow-ice
formation due to the usually warm isothermal temperature profile in the summer ice cover). In order to fill this knowledge
gap, a dedicated program of obtaining much more core sections from the top of the sea ice at different locations would have
to be implemented, with a subsequent crystal fabric and/or oxygen isotope analysis. As indicated above, this is currently not
feasible. The other ice cores taken at ATKA21 and ATKA24 are close to the "typical" sea ice thickness at Atka Bay of 2 m,
and exhibit the expected granular ice at the top from wind and waves, and incorporated platelet ice throughout the rest of the
core. Again, ATKA21 does not show snow-ice formation at the top which might be expected from the snow depth (Figure 4)
and freeboard/water levels (Figure 5) measurements. No evidence from dynamic growth processes is found in these cores.
This is in line with our knowledge so far, especially since the sea ice in that area of the bay typically forms later in the year
and is less influenced by strong winds.

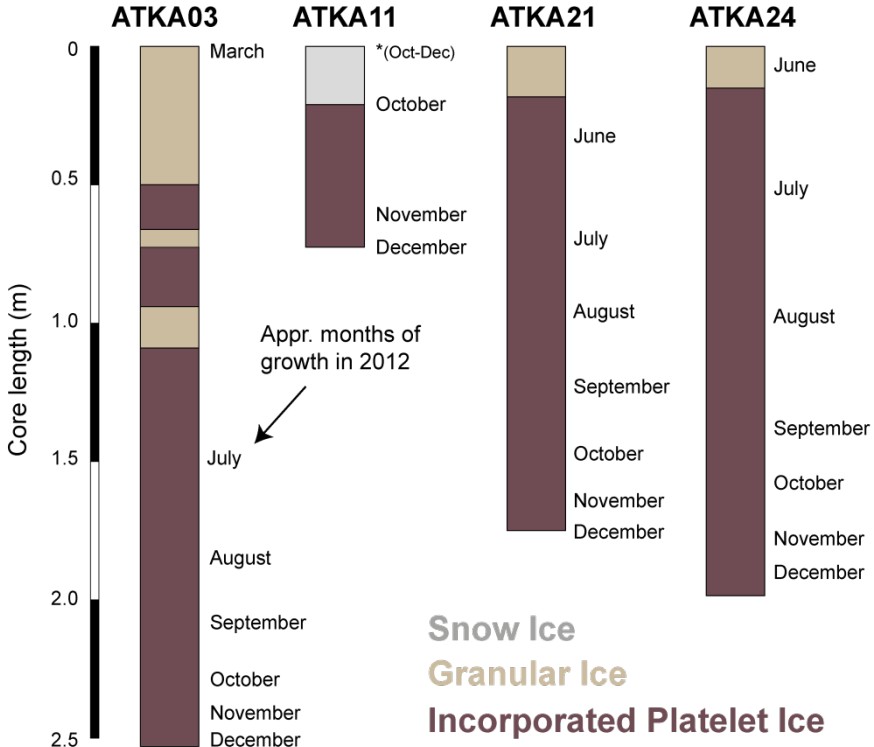

**Figure 9:** Sea-ice crystal fabric from ice cores obtained at four different fast ice sampling sites in December 2012, derived from vertical and horizontal thin sections (0.1m spacing) along the full core length (see also Hoppmann et al., 2015a; Hoppmann et al., 2015b; Hoppmann, 2015).

## 4.5 Implications for multi-disciplinary research

Such a multi-layered, thick sea-ice cover not only very efficiently separates the atmosphere from the ocean with respect to ice growth, but it also influences the exchange of any fluxes between the two climate system components. Thereby, it also strongly impacts the ice-associated ecosystem, which is particularly unique in sub-ice platelet layers (Arrigo, 2014). Günther and Dieckmann (1999) concluded from their study that about 99% of the total fast-ice biomass in Atka Bay originates from algae initially growing in the sub-ice platelet layer. The maximum Chl-a concentration in their study was around 490 mg m$^{-3}$ in the bottom of the fast ice, and 240 mg m$^{-3}$ in the platelet layer in summer, at a site that had up to 0.35 m of snow cover. The authors argued that their total observed fast ice biomass was significantly lower compared to the mostly snow-free fast ice of the Ross Sea. However, it was still in the very upper range of biomass usually found in Antarctic fast ice (Meiners et al., 2018). At the same time, more recent results from 2012 at Atka Bay reveal that Chl-a concentrations can reach up to 900 mg m$^{-3}$ when there is much less snow present (Figure 10).

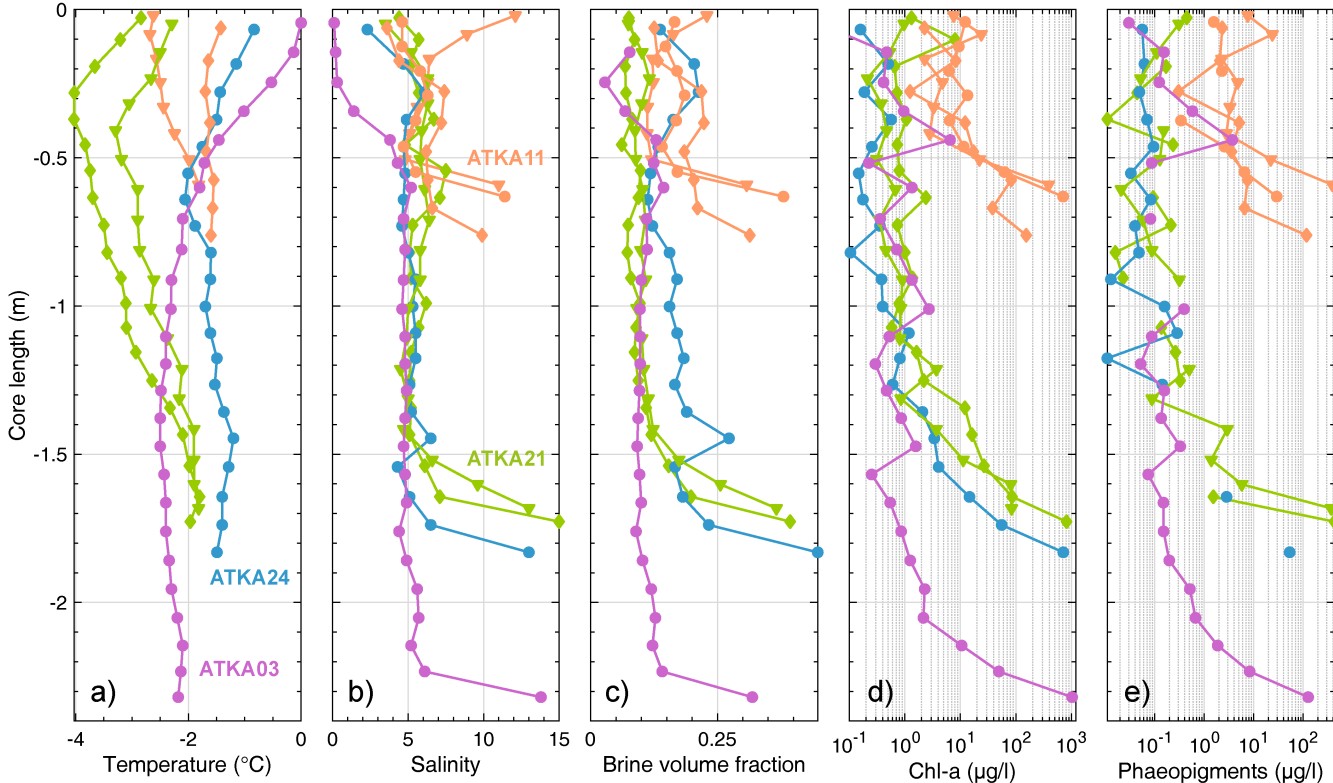

**Figure 10:** Sea-ice physical and biological properties from cores obtained at different fast ice sampling sites in Nov/Dec 2012 (after Hoppmann et al., 2013).

While a few studies exist that investigate shade-adaptation in algae and link algal growth to snow depth on McMurdo Sound fast ice (e.g. Sullivan et al., 1985; McGrath Grossi et al., 1987; Robinson et al., 1995), so far still comparably little is known about the adaptation of the ecosystem in the upper ocean to perennial fast-ice conditions and sub-ice platelet layers. These and similar knowledge gaps that exist with respect to ice-shelf influenced fast-ice regimes can only be addressed by integrated, multi-disciplinary research in comparably easy to access locations in coastal Antarctica, one of which was introduced in this physical study.

## 5 Conclusions

This study presents a unique, 9-year long record (2010 to 2018) of snow depth, freeboard, sea-ice and sub-ice platelet-layer thickness observed at Atka Bay, a coastal Antarctic fast-ice regime in the southeastern Weddell Sea and key region in the Southern Ocean. As one of the longest time series within the Antarctic Fast Ice Network, and complementary to similar records in the Ross Sea (Brett et al., 2020; Langhorne et al., 2015 and references therein), this dataset is expected to serve as an

important baseline in the context of climate change and future sea-ice evolution in this region, and will contribute to an enhanced understanding of the complex interactions between the atmosphere, sea ice, ocean and ice shelves in the Southern Ocean.

For the period of the study presented, and considering individual observations from the 1980s and 1990s, a predominantly seasonal character of the fast-ice regime in Atka Bay is evident without a noticeable trend for any of the analyzed variables. The absence of any trend and the seasonality of surface characteristics associated with the year-round snow cover and negligible surface melting coincides with the prevailing conditions in the Antarctic pack-ice zone. Hence, the described observations in Atka Bay over the last nine years not only allow to document a baseline of the observed parameters, but also to capture processes and properties prior to expected future changes of pack ice in, e.g., the Weddell Sea, due to a changing climate.

Atka Bay is dominated by strong cyclonic events leading to easterly winds which determine not only the freeze-up of the bay in autumn and breakup during summer months, but also govern the year-round snow distribution on the ice. The consequent substantial annual snow accumulation determines both, the magnitude and duration of congelation sea-ice growth, as well as the magnitude and spatial distribution of the frequent negative freeboard and related flooding of the snow/ice interface, and thus potential snow-ice formation. In contrast, platelet ice contributes significantly to the total sea-ice mass balance in this region, both, in its unconsolidated form as an underlying (buoyant) platelet layer, as well as through its incorporation into the solid sea ice (see also Hoppmann et al., 2015a; Hoppmann et al., 2015b). However, our results indicate that, although the platelet layer partly offsets the negative freeboard, it is not buoyant enough to lift the snow/ice interface above sea level against the prevalent weight of the snow.

With regard to the platelet layer and its formation process, we conclude that, although the annual platelet-layer thickness increase of four meters seems to be independent of the age of the fast ice in the bay, the seasonal and inter-annual variability of this layer and thus the associated ocean properties and processes cannot be understood sufficiently by just considering the fast-ice properties alone. We therefore recommend to follow the approach of the New Zealand research program at Scott Base to generally include an oceanographic component into any fast-ice monitoring, especially in regions where ice shelves are present. This combination would allow for quantifying the seasonal interactions between sea ice, ocean and shelf ice even more precisely and thus to better understand current patterns and accumulation rates of platelet ice and associated biomass under the ice as a function of the distance to the shelf-ice and sea-ice edge. These results would provide a solid basis to be applied to all fast-ice areas around Antarctica, and thus make a fundamental contribution to the understanding of the Antarctic climate system.

**Data availability**

All presented meteorological data are archived in PANGAEA at https://doi.pangaea.de/10.1594/PANGAEA.908826. All fast-ice data are archived in PANGAEA at https://doi.pangaea.de/10.1594/PANGAEA.908860 .

## Author contribution

SA conducted most of the analyses for this paper and did the main writing with input from all co-authors. MH contributed the sea ice core data and helped SA to complete the revisions with input from all other authors. MN is the principal investigator of the AFIN work at Neumayer III. MH, MN and SA supervised the sea-ice measurements of the overwintering teams during the study period. MH and SA participated in field campaigns to collect parts of the presented data. HS contributed the meteorological datasets and the related analysis. AF contributed the fast-ice extent dataset and the related analysis.

## Competing interests

The authors declare that they have no conflict of interest.

## Acknowledgements

We are most grateful to the overwintering teams at Neumayer III from 2010 to 2018 for their conducted measurements on the fast ice in Atka Bay. Special thanks are due to the respective meteorologists of the teams who led the sea ice work on site. Also, our work and research at Neumayer III would not have been possible without the extensive support of the AWI logistics. We also acknowledge the scientific support of Christian Haas, the logistical support of Anja Nicolaus, and the technical support of Jan Rohde, all from the Sea Ice Physics section at AWI. This work was supported by the German Research Council (DFG) in the framework of the priority programme ''Antarctic Research with comparative investigations in Arctic ice areas'' by grants to SPP1158, HE2740/12, NI1092/2 and AR1236/1, and the Alfred-Wegener-Institut Helmholtz-Zentrum für Polar- und Meeresforschung. This research was also supported under Australian Research Council's Special Research Initiative for Antarctic Gateway Partnership (Project ID SR140300001). We are grateful to two anonymous reviewers and the editor Jean-Louis Tison for their valuable input, which significantly improved the quality of the presented science.

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
