# Peer review of "Seasonal and interannual variability of landfast sea ice in Atka Bay, Weddell Sea, Antarctica"

_The Cryosphere, 2019_

## Referee Comment (RC1) · Anonymous Referee #1 · 29 Feb 2020

Note: full reference information can be found at the end of this report.

General Comments:

This manuscript presents the results of a semi-continuous 9-year study of sea ice, platelet ice, freeboard and snow conditions in Atka Bay, an embayment in front of the Ekström Ice Shelf located on the coast of Dronning Maud Land in the eastern Weddell Sea, Antarctica. This is a novel data set that is analyzed in the manuscript to elucidate seasonal and interannual variability and determine whether there are any noticeable trends. The results of the analysis indicate that the seasonal character of the fast-ice regime in Atka Bay predominates and no noticeable trends were observed.

The manuscript provides a very valuable dataset for evaluating the fast-ice conditions

in Atka Bay in the context of local and regional atmospheric and oceanic conditions, including the effect of the adjacent Ekström Ice Shelf on the formation of platelet ice. It thus represents an important contribution to the current understanding of how Antarctic fast-ice regimes adjacent to ice shelves are affected by sub-ice-shelf processes, such as the formation of frazil laden Ice Shelf Water plumes.

I think the manuscript represents substantial progress beyond current scientific understanding and merits publication once the comments I have made in the following sections have been addressed. My principal comments are that the authors have not cited in the manuscript a number of studies that have investigated the fast-ice regime in McMurdo Sound, which, similar to Atka Bay, is an area of fast ice growth adjacent to an ice shelf. The inclusion of these studies will, I believe, add greatly to contextualizing and interpreting the data presented in the manuscript. I also would like the authors to review how they use the concept of freeboard in the manuscript, as their definition of freeboard in the text does not align with their Equation 1 and Figure 5.

Specific Comments:

Line 52, I recommend the authors also cite Leonard et al. (2006) as another example of Antarctic landfast sea ice that reached a thickness greater than 2 m and was not perennial.

Line 69. I didn't have Foldvik (1977) in front of me when reviewing the manuscript, but I still feel comfortable enough to question the statement that supercooled ISW favors the formation of floating ice crystals "deep" in the water column. Can the authors define what they mean by "deep" – buoyant ISW needs to rise some distance through the water column to become sufficiently supercooled to initiate frazil crystal formation.

Line 70. The authors have not mentioned other studies that have reported on field observations of frazil laden ice shelf water plumes advecting out from an ice shelf cavity that then rise to the surface, e.g. Mahoney et al. (2011) and Hughes et al. (2014).

[Figure]

Line 71. I suggest the authors include the work of Price et al. (2014) and Brett et al. (2020) as they also report on the accumulation of sub-ice platelet layers under the sea ice.

In Figure 2, what is the uncertainty with the annual fast ice extent estimates? It would be useful to report what the average and standard deviation of the extent was over the study period. Did the authors consider showing fast ice extent anomalies instead of fast ice extents? This would negate the need to repeat the average extent for each year of the study.

Section 2, Line 173. I did not understand what was meant by "An additional metal bar ...". Does that mean the measuring tape had two metal bars at its base? I don't understand how this would work. The authors mention that this is a "modified" thickness tape, but don't describe the characteristics of an "unmodified" thickness tape and hence it is not clear to the reader how the "modified" thickness tape is meant to perform better.

Section 2, Lines 178 – 182. It would be useful for the reader for the authors to explain here why they are both measuring and calculating the freeboard. I believe "Archimedes law" should be "Archimedes principle". Equation 1 is not consistent with the manuscript's statement that a "snow/ice interface above sea-water level is referred to as positive" with respect to freeboard. For example, if there were no snow or platelet ice, Equation 1 would produce a negative freeboard. I assume that the densities of ice and water have been interchanged in line 182 as the density of sea ice cannot be greater than the density of seawater! Authors need to state the "indices" in Equation 1 are thicknesses and should show how the density of the platelet ice is calculated from the ice volume fraction.

On a related note, the authors state that the lines in Figure 5 are "freeboards". They are not "freeboards" (i.e. vertical separation between the water surface and a point of interest, in this case the snow / ice interface). Rather they are measurements and

estimates for where the water surface is relative to the snow/ice interface, with different assumptions (coloured lines). The caption attempts to describe what these lines are, but still incorrectly and confusedly refers to them as freeboards.

The authors generally present what are assumed to be mean values and standard deviations when reporting their observations, but do not explicitly state that this is the case. The authors should confirm what they are presenting, and also, provide an estimate of the uncertainties associated with their measurements.

The authors make a number of references to "thermodynamically grown ice" in the manuscript, but do not provide a definition of what they mean by this, and also do not provide any direct evidence of the mechanism behind sea ice growth, as they have not presented fabric crystal structures as per Hoppmann et al. (2015). At the very least, reference should be made to the Hoppmann et al. study as those measurements were made within the time frame of this study (2010 - 2018) in Atka Bay.

I suggest that the authors re-consider the naming convention they have used for what they term "platelet ice" to help distinguish columnar sea ice from sea ice that has formed by "consolidating" platelet ice. For example, see Hughes et al, 2014, where the term "sub-ice platelet layer" is used to describe the loose platelet crystals under the sea ice, and "consolidated platelet ice" is used to describe that part of the sea ice that formed by congelation ice growing down into the sub-ice platelet layer and consolidating the platelet ice. This approach might also clarify whether the authors are using the term "thermodynamically grown ice" to refer to just columnar ice, or columnar ice and consolidated platelet ice.

I feel this manuscript could benefit greatly from more contextualizing of the results presented here with other studies. For example, in the discussion, there is no mention at all of the body of work on sub-ice platelet layers under fast ice in McMurdo Sound, including both observational and modelling studies. Sub-ice platelet thickness gradients have also been observed (and modelled) in the McMurdo Sound studies. See for

example Hughes et al. (2014), Dempsey et al. (2010), Robinson et al. (2014) and Cheng et al. (2019).

I would also suggest the authors consider producing spatially interpolated plots of the measured variables, similar to what is presented in Price et al. (2014) so that they could better present spatial trends in their data. This would help to strengthen the discussion where they consider mechanisms driving changes in the measured variables, e.g. the "washing-out" mechanism invoked to explain the thinning of platelet ice.

Reference is made in the discussion to oceanographic drivers influencing the platelet ice in Atka Bay, but no oceanographic data is presented or referenced. In particular, reference is made to "underwater topographic features of the ice shelf" potentially leading to "blocking of oceanic circulation patterns", but these features are not shown or cited in other works. Their hypothesis could be strengthened by including supporting oceanographic observations / modelling output.

In Lines 439 – 445 of the Discussion, arguments are put forth for dynamical growth and compaction of the platelet ice layer dominating over thermodynamic growth. Although I agree with the authors that this is probably the case, without fabric crystal structure information, this cannot be said definitively. I also would like the authors to discuss the compaction of platelet ice (referred to as consolidation in other studies) in more detail, as this process has a strong thermodynamic element (refer to Wongpan et al. (2015).

Technical Corrections:

Below is a list of technical corrections / suggestions for the authors to address and consider:

Note: Line numbers over 99 are assumed as the leading digit was cut off in the pre-print.

Section 1: Line 12, change "of fast ice of Atka Bay…" to "of fast ice in Atka Bay…".

Line 15, change ", sea-ice-…" to "and sea-ice".

Line 17, insert a comma after "Neumayer Station" and a second comma after "satellite images". Also insert "us" after "allows".

Line 19, change "meters snow" to "meter thick snow cover".

Line 20, insert "interannual" before "trend".

Line 23, replace "event" with "of landfast sea-ice".

Line 28, replace "on" with "of".

Line 36, replace "extent" with "seaward edge".

Line 54, replace "but also to" with "and".

Line 57, replace "the" with "an".

Line 58, re-arrange "for the Arctic recently" to "recently for the Arctic".

Line 62, replace "particular" with "particularly".

Line 78, replace "in" with "over" and "parts" to "portions".

Line 88, replace "for" with "in".

Line 96, replace "in McMurdo Sound at Scott Base" with "working out of Scott Base in McMurdo Sound" and put "e.g." in front of "Smith et al., 2001) – Smith et al. (2001) is one of the earlier manuscripts but there are many more, such as Smith et al. (2015).

Line 97, replace "Fimbul ice shelf" with "the Fimbul Ice Shelf".

Line 99, replace "the knowledge gap" with "a knowledge gap".

Section 2: Line 112, add "depth" after "275 m".

Line 115, add "the" in front of "sea ice".

Line 117, remove the two semi-colons.
Figure 1a, replace "Grounded ice sheet" with a more accurate description in the caption as this looks to me to be the "land" feature in the Antarctic Digital Database. If it is, add a citation to the Antarctic Digital Database. Add a north arrow.

Figure 1b, add a bounding rectangle to show the extent of Figure 1c.

Line 123 -Figure 1 citation – re-word "in same distance" to improve clarity.

Line 126, replace "the sampling sites" with "adjacent sampling sites" and re-word "all sampled additionally" to improve clarity. Add "is" before "a Sentinel" and provide the required citation for the Sentinel imagery as per https://sentinels.copernicus.eu/documents/247904/690755.

Line 127, delete the "s" from "images"

Line 129, replace "...sea ice, attached" with "...sea ice that is attached"

Line 133, replace the comma with a semi-colon and the add "and" after it.

Line 140, insert "of" before "Atka".

Line 141, replace "currents, winds" with "currents and winds,".

Line 145, replace "causing" with "resulting in".

Line 147, insert "for" in front of "a second time".

Line 149, replace "summer afterwards" with "the following summer".

Line 160, be consistent with spelling of "kilometer" etc. Here it is "kilometer", but the rest of the manuscript uses "...meter".

Line 163, replace "one" with "four more" and replace "5 meter" with "5 meters".

Line 164/165, replace "measurement frequencies" with "the number of observations"

Line 166, re-word the sentence from "however," onwards to improve clarity.
Line 169, replace "Figure 1" with "Figure 1c".

Line 184, replace "we use" with "was used".

Line 192, replace "today" with something like "and beyond the end of the study period".

Line 193, add "sea" in front of "ice".

Line 196, mention that the meteorological data used in this study included 2m air temperatures and 10 m wind velocities.

Line 200, re-word from "The second mode in in the wind..." to improve clarity.

Line 204 or thereabouts, provide some more information on the MET data such as sampling frequency, averaging (if there is any), uncertainties, etc.

Figure 3, x axis title, these are wind speeds, not wind velocities.

Line 207 in Figure 3 caption, again these are wind speeds not wind velocities that are related to the wind directions.

Section 3: Line 219, add "the" in front of "highest snow..." Confirm that accumulations are per year.

Line 223/224. This sentence is confusing. The mean thickness of all of the sea ice in the bay is not varying between 1.74 m, and 2.58 m. These are the means at particular sampling locations. I take it that the mean ice thickness over the entire bay was estimated as being 1.99 m based on the measurements.

Line 229, clarify what is meant by the statement "an average of additional ...." Additional as referenced to what?

Line 230, clarify what is meant by "... increased by another ..."

Line 231, How do the authors know that there is "second-year" platelet ice at these sites? The platelet ice /sub-ice platelet layer is not attached to the overlying sea ice, so could have a different history to the sea ice directly above it. The authors again refer

to "additional" accumulation, how was this determined?

Line 232, add "the" before "highest"

Figure 4, refer to the comment in the Specific Comments section regarding the naming convention for platelet ice.

Figure 4 caption, Line 239, remove "has" before "strongly".

Line 249, correct equation reference to read "Equation 1".

Figure 5, refer to the comments in the Specific Comments section regarding the use of the term freeboard. Correct misspelling "freedboard" in legend.

Line 263, why have the authors referred to "loose platelet ice thickness" here where elsewhere the have used the term "platelet ice"?

Line 277, insert "the" before "highest".

Line 287, how do the authors define "typical thermodynamic sea-ice growth"? 1 m per month sounds very high to me for ice that has only grown "thermodynamically", i.e. by heat being removed from the ocean though the ice to the atmosphere. Are the authors including the growth of consolidated platelet ice as "thermodynamic growth"? If so, this will include ice that has formed within the supercooled ISW.

Line 289, remove "even" in front of "sea-ice melt".

Lines 288 and 289. Line 288 uses "sea-ice thickness rates" and Line 289 uses "melt rate". Suggest authors stick to "thickness rate" and use positive values for growth and negative values for melt. I.e. in Line 289, change "melt rate" to "thickness rate" and change value to a negative.

Figure 6 y axis label. Please change to growth / accumulation rates in m month-1.

Line 303, please clarify what is meant by the statement "the fast-ice thickness over the bay in the south-north and west-east direction is rather constant with . . ." I assume for

the preceding statement that this pertains to the "additional" transect measurements, but this should be made more clear.

Line 304, replace "higher" with "greater".

Line 305, replace "decrease" with "decreases".

Line 306, Figure 7 strictly shows changes in platelet ice layer thickness, not accumulation. This change in thickness can be due to in-situ growth of the platelet ice crystals or the accumulation of new ice crystals flushing out from underneath the ice shelf.

Line 307, add "the" before "lowest".

Line 310, add "the" before "south-to-north".

Figure 7, these plots do not show freeboard, instead they show the height of the water surface relative to the snow/ice interface.

Figure 7 caption, line 315, replace "Overview on" with "Overview of". Line 319 and 320, remove space between "Figure 1" and "c".

Line 329, replace "that" with "the".

Line 333. Re-word "snow cover completely isolates heat fluxes". The heat fluxes are not isolated, rather the snow cover acts as an insulator that reduces the heat fluxes.

Line 336. There are other key studies that should be referenced regarding the consolidation of platelet ice, such as Dempsey et al. (2010).

Line 341, add "of" before "the platelet".

Line 347, replace "Massom et al. (2018) has" with "Massome et al. (2018) have"

Line 352, add a comma after "bay" and replace "strongest" with "strong".

Line 353, add "is observed" after "(Figure 6)". Remove the comma after "both".

Line 354, remove space between "8" and "a".

Line 355, remove "watch".

Line 358, replace "9-year's time series " with "9-year time series".

Line 361. Rates should be expressed as a quantity per unit time, i.e. m year-1.

Line 362, replace "shelfs" with "shelves"; remove "even".

Figure 8a – use units of m to be consistent with other plots. Figure 8a does not show a correlation between 2-meter air temperatures and snow accumulation. Rather it shows a scatter plot comparing these two variables. The authors should include a linear regression if they want to show correlation.

Figure 8 caption. State the dates of the two surveys and whether all sampling sites were included, and confirm that the MET data are averages between these two dates. Replace "Chapter" with "Section". Line 367, add "of" before "fast-ice".

Line 407, this sentence is confusing, please re-word.

Line 412, replace "reveals" with "revealed".

Line 431, replace "plate" with "platelet".

Line 434, replace "caused" with "led to".

Line 446 + 447, this sentence is confusing, please re-word.

Line 453, replace "on" with "of".

Line 479, replace "to quantify" with "for quantifying".

Line 482, remove the space before "ice" in "sea- ice".

Line 488, replace "principle" with "principal".

Line 542, I found online that this publication date should be cited as 2016, but the Polarforschung date is 2015?
References:

Leonard, G. H., C. R. Purdie, P. J. Langhorne, T. G. Haskell, M. J. M.Williams, and R. D. Frew (2006), Observations of platelet ice growth and oceanographic conditions during the winter of 2003 in McMurdo Sound, Antarctica, J. Geophys. Res.,111, C04012, doi:10.1029/2005JC002952.

Dempsey, D. E., Langhorne, P. J., Robinson, N. J., Williams, M. J. M., Haskell, T. G., & Frew, R. D. (2010), Observation and modeling of platelet ice fabric in McMurdo Sound, Antarctica, J. Geophys. Res., 115, C01007, doi:10.1029/2008JC005264.

Mahoney, A. R., A. J. Gough, P. J. Langhorne, N. J. Robinson, C. L. Stevens, M. J. M. Williams, and T. G. Haskell (2011), The seasonal appearance of ice shelf water in coastal Antarctica and its effect on sea ice growth, J. Geophys. Res.,116, C11032, doi:10.1029/2011JC007060.

Hughes, K.G., Langhorne, P.J., Leonard, G.H., Stevens, C.L., (2014), Extension of an Ice Shelf Water plume model beneath sea ice with application in McMurdo Sound, Antarctica, J. Geophys. Res., 119, 8662–8687, doi.org/10.1002/2013JC009411.

Price, D., Rack, W., Langhorne, P. J., Haas, C., Leonard, G., and Barnsdale, K., (2014), The sub-ice platelet layer and its influence on freeboard to thickness conversion of Antarctic sea ice, The Cryosphere, 8, 1031–1039,doi.org/10.5194/tc-8-1031-2014.

Robinson, N. J., Williams, M. J. M., Stevens, C. L., Langhorne, P. J., & Haskell, T. G. (2014), Evolution of a supercooled ice shelf water plume with an actively growing subice platelet matrix, J. Geophys. Res., 119, 3425–3446. doi.org:10.1002/2013JC009399.

Smith, I. J., Gough, A. J., Langhorne, P. J., Mahoney, A. R., Leonard, G. H., Van Hale, R., . . . Haskell, T. G. (2015), First-year land-fast Antarctic sea ice as an archive of ice shelf meltwater fluxes, Cold Regions Science and Technology, 113, 63–70, doi.org/10.1016/j.coldregions.2015.01.007.

Wongpan, P., Langhorne, P. J., Dempsey, D. E., Hahn-Woernle, L. (2015), Simulation of the crystal growth of platelet sea ice with diffusive heat and mass transfer, Annals of Glaciology, 56:69, 127 – 136, doi:10.3189/2015AoG69A777.

Buffo, J. J., Schmidt, B. E., Huber, C. (2018), Multiphase Reactive Transport and Platelet Ice Accretion in the Sea Ice of McMurdo Sound, Antarctica, J. Geophys. Res., 123, 1, (324-345), doi:10.1002/2017JC013345.

Cheng, C., Adrian Jenkins, Paul R. Holland, Zhaomin Wang, Chengyan Liu and Ruibin Xia, (2019), Responses of sub-ice platelet layer thickening rate and frazil-ice concentration to variations in ice-shelf water supercooling in McMurdo Sound, Antarctica, The Cryosphere, 10.5194/tc-13-265-2019, 13, 1, (265-280).

Brett, G.M., Irvin, A., Rack, W., Haas, C., Langhorne, P. J., Leonard, G. H., (2020), Variability in the distribution of fast ice and the sub‐ice platelet layer near McMurdo Ice Shelf, J. Geophys. Res., doi: 10.1029/2019JC015678.

---

## Referee Comment (RC2) · Anonymous Referee #2 · 16 Mar 2020

The paper summarizes a decade of annual in-situ fast-ice observations in Atka Bay, which is the longest and most continuous time series within the Antarctic Fast Ice Network (AFIN). The main dataset is a semi-continuous record of fast-ice thickness, snow depth, freeboard, and sub-ice platelet layer thickness that was collected by overwintering teams between 2010 and 2018. In addition to determining the spatio-temporal variability of the fast-ice cover, this data is co-analyzed with meteorological and oceanographic observations in order to determine how snow and platelet ice influence the local fast-ice mass budget. The discussion at l, 47 p.20 starts with: "In contrast, the relatively thick snow layer on the fast ice in Atka Bay prevents a significant light input to the sea-ice bottom and thus additional biomass production." This statement is speculative considering the absence of light measurements and associated biological production

under the fast ice reported in this study. There is no basis for assuming that the snow cover leads to reduced light and effects on the biological production without specific knowledge of either the light field or the particular organisms that may be, for example, highly shade adapted. Further speculation in subsequent lines that cracks and leads and distance from the ice edge presumes another effect, the increasing light level on the ecosystem. While light may be one component of the algal development in platelet ice, the exchange of seawater with nutrient loads may be even more important for example. As these statements are clearly outside the scope and measurements reported here, I recommend this discussion be deleted. In general, the paper represents a good description of the fast ice features within Atka Bay although lacking some important considerations of the work in McMurdo Sound that the other reviewer has provided more details on. With some modification as outlined here and there, I recommend the paper be published.

Specific Comments

l.27 Replace images with image l. 92 change "today" to 2018/2019 (here and elsewhere) l. 95 delete "a" l.:83 6.2 5m sounds off compared to surrounding values. Is it 0.625 m instead? (See Fig 4 for ATKA11)

l.:88-89 sounds confusing: in summer showing sea ice growth rates increasing, or should it be decreasing? l.05 (105 pg.15?) should decrease be decreases? p.19 l.31 change plate to platelet? p.20 l.47 change concealed to congealed p.20 l-47-54 Heavy algal formation in platelet ice also observed in McMurdo Sound, many km from the light sources mentioned here(ice edge, cracks etc), so organisms may be instead very shade adapted. The strong accumulations also suggest continuous supply of nutrients into the platelet layer so the processes of convective overturning (possibly tidal forcing also) may be responsible for the high algal growth rates in the platelet layer. Experiments by Sullivan and colleagues with varying depths of snow artificially placed on or removed from the ice surface showed that growth proceeded best with an optimal snow depth, rather than no snow. Probably delete this discussion since it is not

supported by measurements or adequate referencing.
* * *

---

## Editor Comment (EC1) · Jean-Louis Tison (Editor) · 30 Mar 2020

Dear Stefanie and Co,

Just as a reminder, the Discussion phase is closed, and both reviewers have recommended publication with detailed suggestions to improve the manuscript. I am therefore waiting for your "author response" with both a "detailed response file" answering all comments of reviewers, and a "revised manuscript file" in two versions: one with highlighted changes, and the other as you suggest to publish it.

Good luck with it,

Jean-Louis Tison

---

## Author Comment (AC1) · 15 May 2020

Response to Anonymous Referee #1

Author comments to a reviewer comment are indicated by #

**preface: the terminology in several figures (legends, axes labels) will need to be updated at a later stage in the review process, since the main author is currently still in the field and file transfers are limited. We already updated several (but not all) figures with regard to the reviewers' other comments, but we decided to not send back and forth all the figures at this point to not stretch the line to the ship. We hope that this is acceptable.**

General Comments: This manuscript presents the results of a semi-continuous 9-year

study of sea ice, platelet ice, freeboard and snow conditions in Atka Bay, an embayment in front of the Ekström Ice Shelf located on the coast of Dronning Maud Land in the eastern Weddell Sea, Antarctica. This is a novel data set that is analyzed in the manuscript to elucidate seasonal and interannual variability and determine whether there are any noticeable trends. The results of the analysis indicate that the seasonal character of the fast-ice regime in Atka Bay predominates and no noticeable trends were observed.

The manuscript provides a very valuable dataset for evaluating the fast-ice conditions in Atka Bay in the context of local and regional atmospheric and oceanic conditions, including the effect of the adjacent Ekström Ice Shelf on the formation of platelet ice. It thus represents an important contribution to the current understanding of how Antarctic fast-ice regimes adjacent to ice shelves are affected by sub-ice-shelf processes, such as the formation of frazil laden Ice Shelf Water plumes. I think the manuscript represents substantial progress beyond current scientific understanding and merits publication once the comments I have made in the following sections have been addressed.

**Dear reviewer, we are very grateful for the effort you have put into this thorough review. We are incredibly happy that you seem to appreciate the work that we have done, and we will do our best to implement your valuable comments into the manuscript to improve it and get it ready for publishing.**

My principal comments are that the authors have not cited in the manuscript a number of studies that have investigated the fast-ice regime in McMurdo Sound, which, similar to Atka Bay, is an area of fast ice growth adjacent to an ice shelf. The inclusion of these studies will, I believe, add greatly to contextualizing and interpreting the data presented in the manuscript.

**We appreciate the comment and we are aware of the wealth of great studies performed in the McMurdo Sound area. We will try to follow the advice and add some of the suggested references where suitable. Having said that, we do think that we should**

still limit the number of McMurdo references since these studies have been compiled pretty thoroughly before, most notably in the Langhorne et al., 2015 paper (which has now been cited a number of times).

I also would like the authors to review how they use the concept of freeboard in the manuscript, as their definition of freeboard in the text does not align with their Equation 1 and Figure 5.

**The concept of freeboard has been reviewed and adjusted based on the reviewer's comments. See also the individual answers below.**

Specific Comments:

Line 52, I recommend the authors also cite Leonard et al. (2006) as another example of Antarctic landfast sea ice that reached a thickness greater than 2 m and was not perennial. #Included as suggested

Line 69. I didn't have Foldvik (1977) in front of me when reviewing the manuscript, but I still feel comfortable enough to question the statement that supercooled ISW favors the formation of floating ice crystals "deep" in the water column. Can the authors define what they mean by "deep" – buoyant ISW needs to rise some distance through the water column to become sufficiently supercooled to initiate frazil crystal formation. #We agree with this comment. We just wanted to express that the crystals form in the water column rather than at the surface, where sea ice usually gorws. Since a more detailed discussing is beyond the scope of this paper, we decided to change the statement to read "... favors the formation of floating ice crystals within the water column (Foldvik, 1977), as opposed to the regular process of sea-ice formation by heat transport from the ocean towards the colder atmosphere."

Line 70. The authors have not mentioned other studies that have reported on field observations of frazil laden ice shelf water plumes advecting out from an ice shelf cavity that then rise to the surface, e.g. Mahoney et al. (2011) and Hughes et al.

(2014). #The suggested references have been included.

Line 71. I suggest the authors include the work of Price et al. (2014) and Brett et al. (2020) as they also report on the accumulation of sub-ice platelet layers under the sea ice. #The suggested references have been included.

In Figure 2, what is the uncertainty with the annual fast ice extent estimates? It would be useful to report what the average and standard deviation of the extent was over the study period.

**We provided a table containing the time series and the corresponding estimated digitization uncertainty as a supplement to this response. We also added the following sentence to the caption: "The average fast-ice extent over the entire time series is 319.2 ± 167.8 kmˆ2, with a mean uncertainty of 86.6 kmˆ2." Please note that, due to the strong annual cycle, the standard deviation is of limited use. The uncertainty estimation methodology is fully detailed in a forthcoming Fraser et al., Earth System Science Datasets paper.**

Did the authors consider showing fast ice extent anomalies instead of fast ice extents? This would negate the need to repeat the average extent for each year of the study.

**We did consider to show the anomalies. We decided against it because the strong seasonal cycle would not be as apparent. I hope the reviewer understands that we prefer to keep the current format.**

Section 2, Line 173. I did not understand what was meant by "An additional metal bar…". Does that mean the measuring tape had two metal bars at its base? I don't understand how this would work. The authors mention that this is a "modified" thickness tape, but don't describe the characteristics of an "unmodified" thickness tape and hence it is not clear to the reader how the "modified" thickness tape is meant to perform better.

**Sorry for the confusion. What was meant is that the regular metal plates usually used**

on the thickness tapes were replaced by a heavy metal bar of about 1-2 kg (and a small rope on one end) in order to penetrate the semi-consolidated platelet layer. This part was changed to read: "Sea-ice and platelet-layer thickness as well as freeboard are measured with a (modified) thickness tape. In order to enable the penetration of the usually semi-consolidated platelet layer, the regular metal plates at the bottom of the thickness tape were replaced by a metal bar of ~2kg. The underside of the platelet layer is determined by gently pulling up the tape and attempting to feel the first resistance to the pulling. Sea-ice thickness was measured either by pulling this modified tape through the entire platelet layer until the solid sea-ice bottom is reached (with a high risk of it getting stuck), or using a regular ice thickness tape. The modified tape is retrieved by pulling a small rope attached to one side of the metal bar."

Section 2, Lines 178 – 182. It would be useful for the reader for the authors to explain here why they are both measuring and calculating the freeboard.

**The freeboard is calculated in order to determine the factors that contribute to flooded/non-flooded regions. In doing so, we neglect within the calculations the thickness of snow or the platelet ice, respectively. To make that clearer to the reader, we changed that part to "In order to determine the major contributors to the measured freeboard (F), we also calculated this parameter . . . ".**

I believe "Archimedes law" should be "Archimedes principle".

**The reviewer is correct and this has been changed as suggested.**

Equation 1 is not consistent with the manuscript's statement that a "snow/ice interface above sea-water level is referred to as positive" with respect to freeboard. For example, if there were no snow or platelet ice, Equation 1 would produce a negative freeboard.

**The reviewer is right. The missing minus sign has been added in front of Equation 1.**

I assume that the densities of ice and water have been interchanged in line 182 as the density of sea ice cannot be greater than the density of seawater!
**The reviewer is right, this has been changed.**

Authors need to state the "indices" in Equation 1 are thicknesses

**We added that the letters I, S and P refer to thicknesses (the indices still refer to the respective medium).**

and should show how the density of the platelet ice is calculated from the ice volume fraction.

**The density of platelet ice is calculated as the product from sea-ice density and ice volume fraction of platelet ice. This has been included in the corresponding paragraph accordingly.**

On a related note, the authors state that the lines in Figure 5 are "freeboards". They are not "freeboards" (i.e. vertical separation between the water surface and a point of interest, in this case the snow / ice interface). Rather they are measurements and estimates for where the water surface is relative to the snow/ice interface, with different assumptions (coloured lines). The caption attempts to describe what these lines are, but still incorrectly and confusedly refers to them as freeboards.

**Thank you for this comment. You are absolutely right. We measured in the field and described in the text of the manuscript what is commonly referred to as "freeboard". Usually a positive freeboard means the water level is below the snow/ice interface and therefore "reverse" to the measurement of snow depth/ice thickness. However, in this figure we thought it is clearer to plot the line with the inverse sign (to display the location of the water level relative to the snow/ice interface). We are not sure whether the reviewer has an issue with this way of display. In any case, we now renamed what was wrongly called "freeboard" due to the opposite sign to "water level". Thus, a (positive) water level equals a negative freeboard. The figure legend and caption have been modified to make this more clear, and we hope that this is acceptable for the reviewer.**

The authors generally present what are assumed to be mean values and standard deviations when reporting their observations, but do not explicitly state that this is the case. The authors should confirm what they are presenting, and also, provide an estimate of the uncertainties associated with their measurements.

**We added a sentence in section 2.3: "Throughout this manuscript, we mainly present the mean values from those up to five single measurements per sampling site."**

The authors make a number of references to "thermodynamically grown ice" in the manuscript, but do not provide a definition of what they mean by this, and also do not provide any direct evidence of the mechanism behind sea ice growth, as they have not presented fabric crystal structures as per Hoppmann et al. (2015). At the very least, reference should be made to the Hoppmann et al. study as those measurements were made within the time frame of this study (2010 - 2018) in Atka Bay.

**We agree that this term has been used inconsistently. We have clarified all instances where "thermodynamically grown ice" is mentioned.**

I suggest that the authors re-consider the naming convention they have used for what they term "platelet ice" to help distinguish columnar sea ice from sea ice that has formed by "consolidating" platelet ice. For example, see Hughes et al, 2014, where the term "sub-ice platelet layer" is used to describe the loose platelet crystals under the sea ice, and "consolidated platelet ice" is used to describe that part of the sea ice that formed by congelation ice growing down into the sub-ice platelet layer and consolidating the platelet ice. This approach might also clarify whether the authors are using the term "thermodynamically grown ice" to refer to just columnar ice, or columnar ice and consolidated platelet ice.

**Thank you for this comment on the terminology. We modified a large number of instances based on your advice that are too many to include here. Please refer to the track changes document.**

[Figure]

I feel this manuscript could benefit greatly from more contextualizing of the results presented here with other studies. For example, in the discussion, there is no mention at all of the body of work on sub-ice platelet layers under fast ice in McMurdo Sound, including both observational and modelling studies. Sub-ice platelet thickness gradients have also been observed (and modelled) in the McMurdo Sound studies. See for example Hughes et al. (2014), Dempsey et al. (2010), Robinson et al. (2014) and Cheng et al. (2019).

**Thank you for this comment. Most of the suggested studies (and some more references) were added to the manuscript.**

I would also suggest the authors consider producing spatially interpolated plots of the measured variables, similar to what is presented in Price et al. (2014) so that they could better present spatial trends in their data. This would help to strengthen the discussion where they consider mechanisms driving changes in the measured variables, e.g. the "washing-out" mechanism invoked to explain the thinning of platelet ice. Reference is made in the discussion to oceanographic drivers influencing the platelet ice in Atka Bay, but no oceanographic data is presented or referenced. In particular, reference is made to "underwater topographic features of the ice shelf" potentially leading to "blocking of oceanic circulation patterns", but these features are not shown or cited in other works. Their hypothesis could be strengthened by including supporting oceanographic observations / modelling output.

**We appreciate the comment on the oceanographic data as a driver of fast-ice evolution and variability. We agree that this is a critical aspect, but we decided to focus on other elements in this particular paper. A separate study focusing on oceanographic observations in this area is currently in preparation. We hope for the reviewer's understanding.**

In Lines 439 – 445 of the Discussion, arguments are put forth for dynamical growth and compaction of the platelet ice layer dominating over thermodynamic growth. Although

[Figure]

I agree with the authors that this is probably the case, without fabric crystal structure information, this cannot be said definitively. I also would like the authors to discuss the compaction of platelet ice (referred to as consolidation in other studies) in more detail, as this process has a strong thermodynamic element (refer to Wongpan et al. (2015).

**this paragraph has been modified extensively. We would love to show ice core crystal structure and associated data from each sampling site each year. And actually this was planned to be a standard component in the monitoring. However, it turned out that this was far too much effort for the wintering teams, which do not have a dedicated sea ice person. All the ice activities are done as a "side job". Same is true for the oceanographic observations, which will from now hopefully be performed despite these issues. The only ice core data we have is from 2011/12, and partly shown in the Hoppmann papers. Since we do not have a consistent fabric crystal structure record, we decided to omit this aspect from this manuscript. Again, we hope for the reviewer's understanding.**

Technical Corrections:

Below is a list of technical corrections / suggestions for the authors to address and consider:

Note: Line numbers over 99 are assumed as the leading digit was cut off in the preprint.

**Sorry or that, not sure what went wrong...**

Section 1: Line 12, change "of fast ice of Atka Bay..." to "of fast ice in Atka Bay...".

**changed as suggested**

Line 15, change ", sea-ice-..." to "and sea-ice".

**changed as suggested**

Line 17, insert a comma after "Neumayer Station" and a second comma after "satellite images". Also insert "us" after "allows".

**changed as suggested**

Line 19, change "meters snow" to "meter thick snow cover".

**changed as suggested**

Line 20, insert "interannual" before "trend".

**changed as suggested**

Line 23, replace "event" with "of landfast sea-ice".

**changed to "of landfast sea ice"**

Line 28, replace "on" with "of".

**changed as suggested**

Line 36, replace "extent" with "seaward edge".

**changed as suggested**

Line 54, replace "but also to" with "and".

**changed as suggested**

Line 57, replace "the" with "an".

**changed as suggested**

Line 58, re-arrange "for the Arctic recently" to "recently for the Arctic".

**changed as suggested**

Line 62, replace "particular" with "particularly".

**changed as suggested**

Line 78, replace "in" with "over" and "parts" to "portions".

**changed as suggested**

Line 88, replace "for" with "in".

**changed as suggested**

Line 96, replace "in McMurdo Sound at Scott Base" with "working out of Scott Base in McMurdo Sound"

**changed as suggested**

and put "e.g." in front of "Smith et al., 2001) – Smith et al. (2001) is one of the earlier manuscripts but there are many more, such as Smith et al. (2015).

**changed to (Langhorne et al., 2015, and references therein) since this seems to give the best comprehensive overview.**

Line 97, replace "Fimbul ice shelf" with "the Fimbul Ice Shelf".

**changed as suggested**

Line 99, replace "the knowledge gap" with "a knowledge gap".

**changed as suggested**

Section 2: Line 112, add "depth" after "275 m".

**changed as suggested**

Line 115, add "the" in front of "sea ice".

**changed to "between the sea ice and the ice-shelf surface"**

Line 117, remove the two semi-colons.

**changed as suggested**

Figure 1a, replace "Grounded ice sheet" with a more accurate description in the caption as this looks to me to be the "land" feature in the Antarctic Digital Database. If it is, add a citation to the Antarctic Digital Database. Add a north arrow.

Interactive
comment

**Reference was added**

Figure 1b, add a bounding rectangle to show the extent of Figure 1c.

**Figure 1b and 1c show the exactly same extent, so a bounding rectangle should not be necessary.**

Line 123 -Figure 1 citation – re-word "in same distance" to improve clarity.

**We removed that part of the sentence since it was made clear in the text that the numbers indicate the distance to the western ice-shelf edge.**

Line 126, replace "the sampling sites" with "adjacent sampling sites"

**changed as suggested**

and reword "all sampled additionally" to improve clarity.

**This has been changed to read "The southern, eastern and western transects were sampled during a field campaign between November and December 2018"**

Add "is" before "a Sentinel"

**Has been changed to "In the background is a Sentinel-1 image taken on December 01, 2018."**

and provide the required citation for the Sentinel imagery as per https://sentinels.copernicus.eu/documents/247904/690755.

**This link seems outdated. The following has been added according to https://asf.alaska.edu/data-sets/sar-data-sets/sentinel-1/sentinel-1-how-to-cite/: "Background: Copernicus Sentinel data 01 December 2018, processed by ESA. "**

Line 127, delete the "s" from "images"

**changed as suggested**
Line 129, replace ". . . sea ice, attached" with ". . . sea ice that is attached"

**changed as suggested**

Line 133, replace the comma with a semi-colon and the add "and" after it.

**changed as suggested**

Line 140, insert "of" before "Atka".

**changed as suggested**

Line 141, replace "currents, winds" with "currents and winds,".

**changed as suggested**

Line 145, replace "causing" with "resulting in".

**changed as suggested**

Line 147, insert "for" in front of "a second time".

**changed as suggested**

Line 149, replace "summer afterwards" with "the following summer".

**changed as suggested**

Line 160, be consistent with spelling of "kilometer" etc. Here it is "kilometer", but the rest of the manuscript uses ". . .meter".

**All instances have been changed to ". . .meter"**

Line 163, replace "one" with "four more" and replace "5 meter" with "5 meters".

**changed as suggested**

Line 164/165, replace "measurement frequencies" with "the number of observations"

**changed as suggested**

[Figure]

Line 166, re-word the sentence from "however," onwards to improve clarity.

**changed to "In years of prevailing second-year ice in the bay (2012/2013, 2014/2015), the number of observations per sampling site may be reduced to one (the center measurement) due to exceptionally thick snow and ice."**

Line 169, replace "Figure 1" with "Figure 1c".

**changed as suggested**

Line 184, replace "we use" with "was used".

**changed as suggested**

Line 192, replace "today" with something like "and beyond the end of the study period".

**changed to "At the meteorological observatory of the nearby wintering station Neumayer III, atmospheric conditions have been recorded since 1981 (König-Langlo and Loose, 2007), in particular covering the sea-ice study period from 2010/2011 onwards, and continuing beyond the end of the study period (Schmithüsen et al., 2019)"**

Line 193, add "sea" in front of "ice".

**changed as suggested**

Line 196, mention that the meteorological data used in this study included 2m air temperatures and 10 m wind velocities.

**changed to read "Therefore, we utilize in this paper the 2m air temperature and the 10m wind velocity data of the meteorological observatory. . ."**

Line 200, re-word from "The second mode in in the wind. . ." to improve clarity.

**Changed to read "The second strongest mode in the wind direction distribution at 270° (westward) is associated to super geostrophic flows . . ."**

Line 204 or thereabouts, provide some more information on the MET data such as

sampling frequency, averaging (if there is any), uncertainties, etc.

**The following paragraph has been added: "The Neumayer data is recorded as minutely averages of typically 10 values per averaging interval. The instrumentation is checked on a daily basis, any erroneous values, e.g. caused by riming or instrument failure, are removed from the record. Therefore, the data quality is high, even though there might be gaps in the records due to the validation routines. Nevertheless, data availability is 99.4% for wind direction, 99.0% for wind speed and 99.7% for temperature data. Uncertainties are essentially those classified by the manufacturers. Instrument details are given in the metadata of the datasets since February 2017 in Schmithüsen et al. (2019), earlier data is documented in König-Langlo and Loose (2007)."**

Figure 3, x axis title, these are wind speeds, not wind velocities.

**The figure has been changed accordingly.**

Line 207 in Figure 3 caption, again these are wind speeds not wind velocities that are related to the wind directions.

**changed as suggested**

Section 3: Line 219, add "the" in front of "highest snow.."

**changed as suggested**

Confirm that accumulations are per year.

**changed to "the highest annual snow accumulation"**

Line 223/224. This sentence is confusing. The mean thickness of all of the sea ice in the bay is not varying between 1.74 m, and 2.58 m. These are the means at particular sampling locations. I take it that the mean ice thickness over the entire bay was estimated as being 1.99 m based on the measurements.

**Has been clarified to read "The mean thermodynamically grown seasonal fast-ice**

thickness based on the measurements during the observation period. . ."

Line 229, clarify what is meant by the statement "an average of additional . . .." Additional as referenced to what?

**Changed to read "In 2013 and 2015, the fast ice in Atka Bay became second-year ice due to blocking icebergs in front of the bay. Within the respective second year, snow depth increased further by an additional 0.88 +- 0.43 in 2013 and 0.74 +- 0.27 m in 2015."**

Line 230, clarify what is meant by ". . .increased by another . . ."

**has been clarified to read "In 2013, the average fast-ice thickness across the bay increased by an additional 1.21 +- 0.42 m, while it even increased by an additional 2.79 m #- 1.48 m in 2015."**

Line 231, How do the authors know that there is "second-year" platelet ice at these sites? The platelet ice /sub-ice platelet layer is not attached to the overlying sea ice, so could have a different history to the sea ice directly above it. The authors again refer to "additional" accumulation, how was this determined?

**has been reformulated to read "In the years of prevalent second year ice in the bay, the thickness of the platelet layer underlying the fast ice increased on average by 5.13 m +- 1.43 m in 2013 (compared to the end of 2012), and 4.11 m +- 1.86 m in 2015 (compared to the end of 2014). During these periods, ATKA11 experienced the highest annual platelet-layer thickness increase of 6.82 m and 6.44 m, respectively."**

Line 232, add "the" before "highest"

**changed as suggested**

Figure 4, refer to the comment in the Specific Comments section regarding the naming convention for platelet ice.

**The figures can only properly be redone after the first author returned from the MO-**

SAiC experiment, which is currently delayed due to the corona crisis. The terminology will be changed after the next review round or during final editing. We hope this is acceptable.

Figure 4 caption, Line 239, remove "has" before "strongly".

**changed as suggested**

Line 249, correct equation reference to read "Equation 1".

**changed as suggested**

Figure 5, refer to the comments in the Specific Comments section regarding the use of the term freeboard. Correct misspelling "freedboard" in legend.

**Typo corrected as suggested**

Line 263, why have the authors referred to "loose platelet ice thickness" here where elsewhere the have used the term "platelet ice"?

**has been corrected to read "platelet-layer thickness"**

Line 277, insert "the" before "highest".

**changed as suggested**

Line 287, how do the authors define "typical thermodynamic sea-ice growth"? 1 m per month sounds very high to me for ice that has only grown "thermodynamically", i.e. by heat being removed from the ocean though the ice to the atmosphere. Are the authors including the growth of consolidated platelet ice as "thermodynamic growth"? If so, this will include ice that has formed within the supercooled ISW.

**Reference to "typical thermodynamic sea-ice growth" has been removed and the text has been modified to read: "The highest average monthly fast-ice growth rates of up to approx. 1 m per month (80th percentile) are measured in autumn, and decrease in the following month until spring. These exceptionally high growth rates result from rapid**

growth of the solid fast ice into the (unconsolidated) sub-ice platelet layer, i.e. from the subsequent freezing of the interstitial water between the platelets in the top part of the platelet layer. In other words, some of the heat within the newly growing ice was already extracted earlier by the ocean during the process of platelet crystal formation in the supercooled Ice Shelf Water plume. "

Line 289, remove "even" in front of "sea-ice melt".

**changed as suggested**

Lines 288 and 289. Line 288 uses "sea-ice thickness rates" and Line 289 uses "melt rate". Suggest authors stick to "thickness rate" and use positive values for growth and negative values for melt. I.e. in Line 289, change "melt rate" to "thickness rate" and change value to a negative.

**changed as suggested**

Figure 6 y axis label. Please change to growth / accumulation rates in m month-1.

**changed as suggested**

Line 303, please clarify what is meant by the statement "the fast-ice thickness over the bay in the south-north and west-east direction is rather constant with . . ." I assume for the preceding statement that this pertains to the "additional" transect measurements, but this should be made more clear.

**has been clarified to read "Considering the thermodynamically grown ice only, the complementary transect data show that . . ."**

Line 304, replace "higher" with "greater".

**changed as suggested**

Line 305, replace "decrease" with "decreases".

**changed as suggested**

Line 306, Figure 7 strictly shows changes in platelet ice layer thickness, not accumulation. This change in thickness can be due to in-situ growth of the platelet ice crystals or the accumulation of new ice crystals flushing out from underneath the ice shelf.

**Good point. The terminology has been changed throughout to read "platelet-layer thickness increase" instead of "accumulation".**

Line 307, add "the" before "lowest".

**changed as suggested**

Line 310, add "the" before "south-to-north".

**changed as suggested**

Figure 7, these plots do not show freeboard, instead they show the height of the water surface relative to the snow/ice interface.

**You are correct. "Freeboard" has been changed to "water level" in this figure.**

Figure 7 caption, line 315, replace "Overview on" with "Overview of".

**changed as suggested**

Line 319 and 320, remove space between "Figure 1" and "c".

**changed as suggested**

Line 329, replace "that" with "the".

**changed as suggested**

Line 333. Re-word "snow cover completely isolates heat fluxes". The heat fluxes are not isolated, rather the snow cover acts as an insulator that reduces the heat fluxes.

**changed to "...until the thickening snow cover more and more reduces the heat flux..."**

Line 336. There are other key studies that should be referenced regarding the consolidation of platelet ice, such as Dempsey et al. (2010).

**This part has been modified to read: "The continuous sea-ice growth (i.e. ocean-atmosphere heat flux) proceeds with decreasing growth rate through fall and winter until the thickening snow cover more and more reduces the heat flux between the upper ocean and the atmosphere, preventing further thermodynamic growth. However, the fast-ice thickness still (albeit very slowly) increases in spring and even during austral summer months. This could potentially be explained by the measurement uncertainty with respect to the large spatial variability of sea-ice thickness even on very small (centimeter) scales, but the consistency in the data suggests that it could also be caused by consolidation processes within the platelet layer below, i.e. in-situ sea-ice growth by heat transport into a supercooled plume residing right beneath the solid fast ice similar to observations in McMurdo Sound (Smith et al., 2012; Leonard et al., 2011; Dempsey et al., 2010; Robinson et al., 2014). So far, in Atka Bay there is only evidence that platelets grow quite large already while still suspended in the water column (Hoppmann et al., 2015b). To what degree an in-situ growth of platelet crystals and consolidation processes that go beyond regular freeze-in of the topmost part of the platelet layer by heat conduction to the atmosphere play a role at Atka Bay still needs to be investigated. In any case, the platelet layer is an efficient buffer between the fast ice and the incoming warmer water in summer (Eicken and Lange, 1989), so the lack of noticeable fast-ice bottom melt is expected. Oceanographic (winter) data is sparse, and the monitoring at Atka Bay has recently been extended to also include regular CTD casts, whenever the (challenging) conditions and time constraints allow. An analysis of available CTD data in Atka Bay is currently ongoing, and will be shown in a future dedicated study."**

Line 341, add "of" before "the platelet".

**changed as suggested**

Line 347, replace "Massom et al. (2018) has" with "Massome et al. (2018) have"

**changed as suggested**

Line 352, add a comma after "bay" and replace "strongest" with "strong".

**changed as suggested**

Line 353, add "is observed" after "(Figure 6)". Remove the comma after "both".

**changed as suggested**

Line 354, remove space between "8" and "a".

**changed as suggested**

Line 355, remove "watch".

**changed as suggested**

Line 358, replace "9-year's time series " with "9-year time series".

**changed as suggested**

Line 361. Rates should be expressed as a quantity per unit time, i.e. m year-1.

**changed as suggested**

Line 362, replace "shelfs" with "shelves"; remove "even".

**changed as suggested**

Figure 8a – use units of m to be consistent with other plots. Figure 8a does not show a correlation between 2-meter air temperatures and snow accumulation. Rather it shows a scatter plot comparing these two variables. The authors should include a linear regression if they want to show correlation.

**Units have been changed to m. We included a linear regression now.**

Figure 8 caption. State the dates of the two surveys and whether all sampling sites were included, and confirm that the MET data are averages between these two dates.

**This figure includes measurements throughout the entire analysis period (2010-2018) as well as from all sampling sites. To clarify that, we changed the caption to read: "Scatter plot comparing (a) the average 2-meter air temperature (see Section 2.4) and the snow accumulation between two consecutive surveys, and (b) increasing (positive values) and decreasing (negative values) fast-ice extent and platelet-layer thickness between two consecutive surveys. The analysis includes all measurements at all sampling sites throughout the study period from 2010 to 2018. Blue circles and red crossed denote the respective mean and maximum values within the time frame between the consecutive measurements. Colored solid lines in Figure (b) show the linear regression between both parameters with the respective correlation coefficients R."**

Replace "Chapter" with "Section".

**changed as suggested**

Line 367, add "of" before "fast-ice".

**not changed**

Line 407, this sentence is confusing, please re-word.

**has been changed to read "Examining the spatial distribution of snow over the bay, the considerably lower snow depth at ATKA24 compared to all other sampling sites is striking, and most likely related to the proximity to the ice-shelf edge in approximately 1 km distance."**

Line 412, replace "reveals" with "revealed".

**changed as suggested**

Line 431, replace "plate" with "platelet".

**changed to "platelet layer"**

Line 434, replace "caused" with "led to".

**changed as suggested**

Line 446 + 447, this sentence is confusing, please re-word.

**This has been changed to read "Considering the large sea-ice thickness of around 2 m, as well as the insulating effect of the thick snow cover on top, the contribution of heat conduction to the atmosphere to sea-ice growth is likely very limited. Instead, it is highly likely that dynamical growth as well as growth related to the consolidation of the platelet layer dominates the thickening of the perennial fast ice, . . .". We hope that this is more clear.**

Line 453, replace "on" with "of".

**changed as suggested**

Line 479, replace "to quantify" with "for quantifying".

**changed as suggested**

Line 482, remove the space before "ice" in "sea- ice".

**changed as suggested**

Line 488, replace "principle" with "principal".

**changed as suggested**

Line 542, I found online that this publication date should be cited as 2016, but the Polarforschung date is 2015?

**The year has been changed to 2016.**

References:

Leonard, G. H., C. R. Purdie, P. J. Langhorne, T. G. Haskell, M. J. M.Williams, and R. D. Frew (2006), Observations of platelet ice growth and oceanographic conditions during the winter of 2003 in McMurdo Sound, Antarctica, J. Geophys. Res.,111, C04012, doi:10.1029/2005JC002952.

**This reference has been included.**

Dempsey, D. E., Langhorne, P. J., Robinson, N. J., Williams, M. J. M., Haskell, T. G., & Frew, R. D. (2010), Observation and modeling of platelet ice fabric in McMurdo Sound, Antarctica, J. Geophys. Res., 115, C01007, doi:10.1029/2008JC005264.

**This reference has been included.**

Mahoney, A. R., A. J. Gough, P. J. Langhorne, N. J. Robinson, C. L. Stevens, M. J. M. Williams, and T. G. Haskell (2011), The seasonal appearance of ice shelf water in coastal Antarctica and its effect on sea ice growth, J. Geophys. Res.,116, C11032, doi:10.1029/2011JC007060.

**This reference has been included.**

Hughes, K.G., Langhorne, P.J., Leonard, G.H., Stevens, C.L., (2014), Extension of an Ice Shelf Water plume model beneath sea ice with application in McMurdo Sound, Antarctica, J. Geophys. Res., 119, 8662–8687, doi.org/10.1002/2013JC009411.

**This reference has been included.**

Price, D., Rack, W., Langhorne, P. J., Haas, C., Leonard, G., and Barnsdale, K., (2014), The sub-ice platelet layer and its influence on freeboard to thickness conversion of Antarctic sea ice, The Cryosphere, 8, 1031–1039,doi.org/10.5194/tc-8-1031-2014.

**This reference has been included.**

Robinson, N. J., Williams, M. J. M., Stevens, C. L., Langhorne, P. J., & Haskell, T. G. (2014), Evolution of a supercooled ice shelf water plume with an actively growing subice platelet matrix, J. Geophys. Res., 119, 3425–3446.

doi.org:10.1002/2013JC009399.

**This reference has been included.**

Brett, G.M., Irvin, A., Rack, W., Haas, C., Langhorne, P. J., Leonard, G. H., (2020), Variability in the distribution of fast ice and the subâĚŸARĔĞ ice platelet layer near McMurdo Ice Shelf, J. Geophys. Res., doi: 10.1029/2019JC015678.

**This reference has been included.**

**We decided to not include the following suggested references:**

Wongpan, P., Langhorne, P. J., Dempsey, D. E., Hahn-Woernle, L. (2015), Simulation of the crystal growth of platelet sea ice with diffusive heat and mass transfer, Annals of Glaciology, 56:69, 127 – 136, doi:10.3189/2015AoG69A777.

Buffo, J. J., Schmidt, B. E., Huber, C. (2018), Multiphase Reactive Transport and Platelet Ice Accretion in the Sea Ice of McMurdo Sound, Antarctica, J. Geophys. Res., 123, 1, (324-345), doi:10.1002/2017JC013345.

Smith, I. J., Gough, A. J., Langhorne, P. J., Mahoney, A. R., Leonard, G. H., Van Hale, R., .. Haskell, T. G. (2015), First-year land-fast Antarctic sea ice as an archive of ice shelf meltwater fluxes, Cold Regions Science and Technology, 113, 63–70, doi.org/10.1016/j.coldregions.2015.01.007.

Cheng, C., Adrian Jenkins, Paul R. Holland, Zhaomin Wang, Chengyan Liu and Ruibin Xia, (2019), Responses of sub-ice platelet layer thickening rate and frazil-ice concentration to variations in ice-shelf water supercooling in McMurdo Sound, Antarctica, The Cryosphere, 10.5194/tc-13-265-2019, 13, 1, (265-280).

Please also note the supplement to this comment:
https://www.the-cryosphere-discuss.net/tc-2019-293/tc-2019-293-AC1-supplement.pdf

---

## Author Comment (AC2) · 15 May 2020

Note: Author comment to a reviewer comment is indicated by #

The paper summarizes a decade of annual in-situ fast-ice observations in Atka Bay, which is the longest and most continuous time series within the Antarctic Fast Ice Network (AFIN).

The main dataset is a semi-continuous record of fast-ice thickness, snow depth, freeboard, and sub-ice platelet layer thickness that was collected by overwintering teams between 2010 and 2018. In addition to determining the spatio-temporal variability of the fast-ice cover, this data is co-analyzed with meteorological and oceanographic observations in order to determine how snow and platelet ice influence the local fast-ice

mass budget.

The discussion at l, 47 p.20 starts with: "In contrast, the relatively thick snow layer on the fast ice in Atka Bay prevents a significant light input to the sea-ice bottom and thus additional biomass production." This statement is speculative considering the absence of light measurements and associated biological production under the fast ice reported in this study. There is no basis for assuming that the snow cover leads to reduced light and effects on the biological production without specific knowledge of either the light field or the particular organisms that may be, for example, highly shade adapted. Further speculation in subsequent lines that cracks and leads and distance from the ice edge presumes another effect, the increasing light level on the ecosystem. While light may be one component of the algal development in platelet ice, the exchange of seawater with nutrient loads may be even more important for example. As these statements are clearly outside the scope and measurements reported here, I recommend this discussion be deleted.

**The reviewer is correct. We are (obviously) no experts on fast-ice biology, and this is actually exactly the reason we brought aspect this up. The point we wanted to make here is just that there are still so many knowledge gaps with respect to the linkages between the physical platelet-layer system (and fast ice in general) and the uniquely adapted ecosystem, and that these can only be addressed by a multi-disciplinary approach in a handful of accessible ice-shelf influenced fast ice locations. While this has been achieved to some degree in McMurdo Sound in the past decades, the authors have been working towards such an integrative research program in Atka Bay for years, but it is quite hard to get the different groups together and the necessary funding. This paper can be considered as a description of the physical system (which the reviewer also recognized in his statement above), and it should be complemented by an ecosystem study (which is currently not happening unfortunately). In order to make this intent more clear, we would really like to keep this aspect and changed this paragraph to read**

"4.4 Implications for multi-disciplinary research Such a multi-layered, thick ice cover not only very efficiently separates the atmosphere from the ocean with respect to ice growth, but it also influences the exchange of any fluxes between the two climate system components. Thereby, it also strongly impacts the ice-associated ecosystem, which is particularly unique in sub-ice platelet layers (Arrigo, 2014). From Günther and Dieckmann (1999) it is known that about 99% of the (substantial) total fast-ice biomass in Atka Bay originates from algae being attached to the platelets that congealed to the fast-ice bottom. This is particularly interesting because a thick snow layer as present on the fast ice in Atka Bay prevents a significant light input to the sea ice – ocean interface, and is usually expected to limit biomass production. While a few studies exist that investigate shade-adaptation in algae and link algal growth to snow depth on McMurdo Sound fast ice (e.g. Sullivan et al., 1985; McGrath Grossi et al., 1987; Robinson et al., 1995), so far still comparably little is known about the adaptation of the ecosystem in the upper ocean to perennial fast-ice conditions and sub-ice platelet layers. These and similar knowledge gaps that exist with respect to ice-shelf influenced fast-ice regimes can only be addressed by integrated, multi-disciplinary research in comparably easy to access locations in coastal Antarctica, one of which was introduced in this study." In general, the paper represents a good description of the fast ice features within Atka Bay although lacking some important considerations of the work in McMurdo Sound that the other reviewer has provided more details on. With some modification as outlined here and there, I recommend the paper be published.

**We hope that we were able to address all the comments and concerns of the other reviewer to his/her satisfaction. We also hope that these modifications will also be acceptable to this reviewer.**

Specific Comments

l.27 Replace images with image

**changed as suggested**

l. 92 change "today" to 2018/2019 (here and elsewhere)

**has been changed to read "in particular covering the sea-ice study period from 2010/2011 onwards, and continuing beyond the end of the study period".**

**Another instance was changed to "so far".**

l. 95 delete "a"

**changed as suggested**

l.:83 6.2 5m sounds off compared to surrounding values. Is it 0.625 m instead? (See Fig 4 for ATKA11)

**While you are right that this value seems off, this is what was reported by the team in the field. Of course it could be some sort of measurement error, but we do not want to just remove it from the data at this point. This part was modified to read: "The maximum decrease of 6.25 m per month occurred at ATKA11 in 2013 (80th percentile). However, it is highly likely that this is a measurement error."**

l.:88-89 sounds confusing: in summer showing sea ice growth rates increasing, or should it be decreasing?

**this has also been mentioned by reviewer 1 and was clarified.**

l.05 (105 pg.15?) should decrease be decreases?

**changed as suggested**

p.19 l.31 change plate to platelet?

**changed to "platelet layer"**

p.20 l.47 change concealed to congealed

**changed as suggested**

p.20 l-47-54 Heavy algal formation in platelet ice also observed in McMurdo Sound,

many km from the light sources mentioned here(ice edge, cracks etc), so organisms may be instead very shade adapted. The strong accumulations also suggest continuous supply of nutrients into the platelet layer so the processes of convective overturning (possibly tidal forcing also) may be responsible for the high algal growth rates in the platelet layer. Experiments by Sullivan and colleagues with varying depths of snow artificially placed on or removed from the ice surface showed that growth proceeded best with an optimal snow depth, rather than no snow. Probably delete this discussion since it is not supported by measurements or adequate referencing

**The reviewer is right. The intent was not to provide wild speculations, but rather to highlight the need for a multi-disciplinary research program beyond McMurdo Sound (see above answer).**

Please also note the supplement to this comment:
https://www.the-cryosphere-discuss.net/tc-2019-293/tc-2019-293-AC2-supplement.pdf
* * *

---

## Author Response (AR1)

**Editor Decision: Publish subject to minor revisions (review by editor)** (26 May 2020) by Jean-Louis Tison

Comments to the Author:
Hi Stefanie, Mario and co-authors,

I have now read your detailed responses to the two reviewers, but, if I am not mistaken, could not find the file with revised manuscript, highlighted with changes. I guess you were waiting for the changes to the figures, with understandable reasons.

Response: We are sorry for the missing manuscript file. It could have been provided easily, but it was not obvious in the web interface where it should have been uploaded. Even in the documentation about the review process it was mentioned that the revised file was only supposed to be uploaded after the editor gave his/her ok to the responses. Maybe it was a simply oversight on our end, but the journal could consider to make it more obvious where to upload that file.

I believe you have answered most of the comments from the reviewer. I am still a little bit uneased by missing any data on ecosystem and ocean circulation, as these items were underlined both by the two reviewers and by myself initially.

Response: While we still believe that the combination of the unique in-situ drilling dataset with remote sensing and meteorology observations makes a sound story, we also understand your and the reviewer's concerns. We took this issue seriously, and in our present (major) revision, we tried our best to mitigate the unease about the lack of ecosystem data, oceanographic data and also sea ice crystal fabric mentioned in one of your other comments. Please note that there is no ecosystem program running at the station, so there are unfortunately no regular observations available. It has led to such desperation that the authors were doing some biological sampling (in this case, chl-a filtration of a few cores in one season) themselves. Sea ice coring was initially planned to be a regular activity, but was logistically not feasible. Oceanographic data (such as shown in Hoppmann 2015) is in principle available, but needs more analysis and even then might not give a whole picture about circulation patterns etc. We have been trying to push for more oceanographic observations, but, as with coring, is currently not feasible (yet). In short, we just don't have the suitable datasets at hand right now to provide a comprehensive, overarching picture. We understand this (physical) paper as a foundation for further investigations. A point of the paper is to highlight that such data is urgently needed to fully understand this system. We hope for your understanding.

Having said all that, we have decided to add some entirely new figures and paragraphs regarding the above-mentioned aspects to the manuscript, which hopefully help to support the main points and convince you to accept it for publication in your journal.

The following paragraph about sea ice crystal fabric has been added to the discussion:

**4.4 Sea-ice growth history**

A detailed study of sea ice crystal fabric by means of visual inspection of thick/thin sections or with the help of an automated fabric analyzer can help immensely to determine the dominant growth processes in a given area of interest. At the same time, the growth history of fast ice is to a large degree governed by the timing of the formation of a persistent ice cover, and can only be interpreted accurately by the help of as much auxiliary information as possible, most importantly from regular satellite imagery such as MODIS, Sentinel-1 or Radarsat.

It has been planned since the start of the AFIN monitoring at Atka Bay in 2010 to regularly obtain sea ice cores for crystal fabric analysis. A set of cores from the six main sampling sites (Figure 1) has been obtained in 2011, and again in 2012. Only 4 out of these 12 cores have been processed so far (all from 2012), which is obviously only a very small sample size compared to the decade of measurements shown above. While the limited ice core data thereby is insufficient to make general statements about sea ice growth processes at Atka Bay, we provide this data here to highlight a few major aspects, some of which have already been discussed earlier.

[Figure]

**Figure 9:** Sea-ice crystal fabric from ice cores obtained at four different fast ice sampling sites in December 2012, derived from vertical and horizontal thin sections (0.1m spacing) along the full core length (see also Hoppmann et al., 2015a; Hoppmann et al., 2015b; Hoppmann, 2015).

From the (limited) data we have from the four 2012 cores (Figure 9), it is evident that 1. there is no columnar texture at all; 2. there is a small fraction of granular ice in the top parts of three cores; 3. there is a small fraction of snow ice in one core and 4. all cores are dominated by incorporated platelet ice. The core from the western part of Atka Bay (ATKA03) exhibits a comparably high fraction of granular ice: a 0.5m long section at the top, and 2 smaller sections a little bit deeper, with some incorporated platelet ice in between. This crystal fabric is a manifestation of the dynamic conditions under which the initial growth takes place, and supports the other datasets shown above. The strong

easterly winds (Figure 3) keep pushing the initially forming thin ice towards the western ice shelf edge, which leads to a grinding of the fragile frazil crystals, and subsequently to a rafting of the newly formed ice. This process seems to be still relevant even after the ice has thickened to >0.5 m, probably by very strong winds. In this way, the thickening rate of the sea ice is greatly accelerated initially (Figure 4). The absence of exclusively columnar ice is evidence that there are already platelet crystals emerging from the cavity very early in the season. While it has been suggested in an earlier study that such crystals would be present in the bay from June onwards (Hoppmann et al., 2015b), there is a possibility that they might arrive even earlier, at least in parts of the bay close to the outflow of ISW. While the ice core taken at ATKA11 is not representative at all for sea ice in the bay due to an early breakup event and subsequent late refreezing, the presence of snow ice is an evidence for a process that we would argue plays an underestimated role in this region. However, we currently do not have any more direct evidence for the wide presence of snow ice at Atka Bay (due to the lack of ice core data) other than the observations of negative freeboard in our main dataset (Figure 5), and several observations of extensive surface flooding from summer campaigns.  In order to fill this knowledge gap, a dedicated program of obtaining much more core sections from the top of the sea ice at different locations would have to be implemented, with a subsequent crystal fabric and/or oxygen isotope analysis. As indicated above, this is currently not feasible. The other ice cores taken at ATKA21 and ATKA24 are close to the "typical" sea ice thickness at Atka Bay of 2 m, and exhibit the expected granular ice at the top from wind and waves, and incorporated platelet ice throughout the rest of the core. No evidence from dynamic growth processes is found in these cores. This is in line with our knowledge so far, especially since the sea ice in that area of the bay typically forms later in the year and is less influenced by strong winds.

The following paragraph about multi-disciplinary research (including the requested ecosystem data) has been added to the discussion:

**4.5 Implications for multi-disciplinary research**

Such a multi-layered, thick sea-ice cover not only very efficiently separates the atmosphere from the ocean with respect to ice growth, but it also influences the exchange of any fluxes between the two climate system components. Thereby, it also strongly impacts the ice-associated ecosystem, which is particularly unique in sub-ice platelet layers (Arrigo, 2014). Günther and Dieckmann (1999) concluded from their study that about 99% of the total fast-ice biomass in Atka Bay originates from algae initially growing in the sub-ice platelet layer. The maximum Chl-a concentration in their study was around 490 mg m$^{-3}$ in the bottom of the fast ice, and 240 mg m$^{-3}$ in the platelet layer in summer, at a site that had up to 0.35 m of snow cover. The authors argued that their total observed fast ice biomass was significantly lower compared to the mostly snow-free fast ice of the Ross Sea. However, it was still on the very upper range of biomass usually found in Antarctic fast ice (Meiners et al., 2018). At the same time, more recent results from 2012 reveal that Chl-a concentrations can reach up to 900 mg m$^{-3}$ when there is much less snow present (Fig. 9).

[Figure]

**Figure 10:** Sea-ice physical and biological properties from cores obtained at different fast ice sampling sites in Nov/Dec 2012 (after Hoppmann et al., 2013).

While a few studies exist that investigate shade-adaptation in algae and link algal growth to snow depth on McMurdo Sound fast ice (e.g. Sullivan et al., 1985; McGrath Grossi et al., 1987; Robinson et al., 1995), so far still comparably little is known about the adaptation of the ecosystem in the upper ocean to perennial fast-ice conditions and sub-ice platelet layers. These and similar knowledge gaps that exist with respect to ice-shelf influenced fast-ice regimes can only be addressed by integrated, multi-disciplinary research in comparably easy to access locations in coastal Antarctica, one of which was introduced in this physical study.

**Finally, we added the following paragraph about oceanographic data into the discussion (section 4.2):**

Regarding the properties of the ocean in this region with respect to its interaction with the ice shelf and sea ice, Hoppmann et al., 2015b used a subset of oceanographic data collected by the nearby PALAOA hydrographic observatory (Boebel et al., 2006) to link fast ice observations to ocean properties. A more recent study by Smith et al. (2020) helped to constrain the boundary conditions for Ice Shelf Water outflow by mapping in great detail the cavity geometry of the Ekström Ice Shelf. This study also shows data from repeated CTD casts through a borehole in the ice shelf, revealing the buoyant outflow of Ice Shelf Water in a relatively shallow surface layer. While these efforts help to better understand the complex system of ice shelf-ocean-sea ice interaction in this region, we conclude that a more comprehensive, year-round oceanographic study that also implements a dedicated survey program is urgently needed as a complement to the sea ice monitoring in order to investigate in more detail the outflow of Ice Shelf Water and the complex processes involved in the redistribution of platelet crystals that emerge from the ice shelf cavity.

I would therefore like you to consider the following further comments in your response:

a) Comment of reviewer 1 on "contextualizing" the discussion of the results with the observation and modelling work Hughes, Dempsey, Robinson and Cheng: it is good that you have included those refrences, but, not having the new version of the manuscript, I cannot judge on how you adequately used these references to "contextualize". Please be sure this explicitely comes out in your final manuscript version

Response: We hope that the updated manuscript addresses your concerns.

b) If the process of "blocking by underwater topographic features" is invoked, it has to be documented one way or another.

Response: Ice rises have now been mentioned explicitly, and references to Hoppmann, 2015 and Figure 1 have been made, where their presence can be seen in the radar signature.

Also, I can understand that you would like to expand on the oceanographic part in a companion paper, but at least mention it as "work under way" , "in prep", "submitted" or "in press"...

Response: see above. We prefer not to "overpromise", and decided to omit a statement that work is on the way.

c) Fabric and crystal structure: please, at least clearly acknowledge their interest and the unavailibilty for logistic reason, or eventually use infos from Hoppman papers..

Response: See above

d) Reviewer 2 has underlined my comments on the total absence of data on ecosystem and related variables. He asked for delete. I can understand you want to show how your physical data set would be of interest for future multidisciplinary work on this...I would therefore accept your comment, but with two wording changes:
- change "...prevents a significant light input..." to "... should prevent..."

Response: This part has been removed. While we do even have some spectral irradiance data from within the platelet layer available along with the Chl-a measurements above, we consider this far beyond the scope of this paper. Again, the ecosystem part that was majorly criticized especially by reviewer #2 was only meant to highlight a significant gap in the current research at Atka Bay. We hope this has been made clearer now.

- change: "...which was introduced in this study..." to "...which was introduced in this physical study"...

Response: Has been changed as suggested.

-ensure that there is no discussion left on the potential role of leads and cracks on the light regime or other matters that have not been documented...

Response: The discussion has been adjusted and the speculative aspects have been removed or reformulated.

e) My initial comments on "ocean circulation" and "ecosystem" have been adressed above. There is still my comment on "snow ice formation" which is mentioned several time without any data...the same stands for this...did you document it with isotopic measurements (since there is no crystallography)?...If it is just hypothetical, don't mention snow ice formation in the manuscript.

Response: We did not do any isotope analysis. We now show evidence of snow ice in the section about sea ice growth. The following text has been added to the new sea ice growth history part (see above):

While the ice core taken at ATKA11 is not representative at all for sea ice in the bay due to an early breakup event and subsequent late refreezing, the presence of snow ice is an evidence for a process that we would argue plays an underestimated role in this region. However, we currently do not have any more direct evidence for the wide presence of snow ice at Atka Bay (due to the lack of ice core data) other than the observations of negative freeboard in our main dataset (Figure 5), and several observations of extensive surface flooding from summer campaigns. In order to fill this knowledge gap, a dedicated program of obtaining much more core sections from the top of the sea ice at different locations would have to be implemented, with a subsequent crystal fabric and/or oxygen isotope analysis. As indicated above, this is currently not feasible.

---

## Editor Decision (ED1)

**Seasonal and interannual variability of landfast sea ice in Atka Bay, Weddell Sea, Antarctica**

Stefanie Arndt[1], Mario Hoppmann[1], Holger Schmithüsen[1], Alexander D. Fraser[2,3], Marcel Nicolaus[1]

[1]Alfred-Wegener-Institut Helmholtz-Zentrum für Polar- und Meeresforschung, 27570 Bremerhaven, Germany

[2] Institute for Marine and Antarctic Studies, University of Tasmania, Hobart 7001, Tasmania, Australia

[3]Antarctic Climate & Ecosystems Cooperative Research Centre, University of Tasmania, Hobart 7001, Tasmania, Australia

*Correspondence to*: Stefanie Arndt (stefanie.arndt@awi.de)

**Abstract.** Landfast sea ice (fast ice) attached to Antarctic (near-)coastal elements is a critical  component of the local physical and ecological systems. Through its direct coupling with the atmosphere and ocean, fast ice properties are also a potential indicator of processes related to a changing climate. However, in-situ fast-ice observations in Antarctica are extremely sparse because of logistical challenges and harsh environmental conditions. Since 2010, a monitoring program observing the seasonal evolution of fast ice in Atka Bay has been conducted as part of the Antarctic Fast Ice Network (AFIN). The bay is located on the north-eastern edge of Ekström Ice Shelf in the eastern Weddell Sea, close to the German wintering station Neumayer III. A number of sampling sites have been regularly revisited each year between annual ice formation and breakup  to obtain a continuous record of sea-ice and sub-ice platelet-layer thickness, as well as snow depth and freeboard across the bay.

Here, we present the time series of these measurements over the last nine years. Combining  these with observations from the nearby Neumayer III meteorological observatory as well as auxiliary satellite images enables us to relate the seasonal and interannual fast-ice cycle to the factors that influence  their evolution.

On average, the annual consolidated fast-ice thickness at the end of the growth season is about two meters, with a loose platelet layer of four meter thickness beneath, and 0.70 meter thick snow on top. Results highlight the predominately seasonal character of the fast-ice regime in Atka Bay without a significant interannual trend in any of the observed variables over the nine-year observation period. Also, no changes are evident when comparing with sporadic measurements in the 1980s and 90s. It is shown that  strong easterly winds in the area govern the year-round snow redistribution and also trigger the breakup of fast ice in the bay during summer months.

Due to the substantial snow accumulation on the fast ice, a characteristic feature is frequent negative freeboard, associated flooding of the snow/ice interface, and a likely subsequent snow ice formation. The buoyant platelet layer beneath negates the snow weight to some extent, but snow thermodynamics is identified as the main driver of the energy and mass budgets for the fast-ice cover in Atka Bay.

**Commented [TJ1]:** Here, and at many other places, it is mentioned "snow redistribution" .. this term is very specific and refers to the reworking of initial snow deposition…I don't think this paper discusses those processes at all… I would stick to "snow distribution" everywhere..

[revised manuscript text omitted]

**Commented [TJ5]:** There probably is internal melt favoring instability and break up..

**Commented [TJ6]:** Ambiguous…I would delete this…There are peaks in 2010-2011, with similar features in 2001, 2002, 2015.. To me there is only one exception.. and that is 2012-2013…this range of years is therefore a strange choice to me…

**Commented [TJ7]:** Not very different from 2000-2001!.. do we have an impact of icebergs there too?..
And 2008-2009 are even higher than average than 2014-2015..

[revised manuscript text omitted]

**Commented [TJ18]:** Could this be shown also on the Figure 1 of this paper by a specific symbol? Line of surface fractures of the ice shelf? Summit of ice rise?...

outflow by mapping in great detail the cavity geometry of the Ekström Ice Shelf. This study also shows data from repeated
CTD casts through a borehole in the ice shelf, revealing the buoyant outflow of Ice Shelf Water in a relatively shallow
surface layer. While these efforts help to better understand the complex system of ice shelf-ocean-sea ice interaction in this
region, we  suggest that a more comprehensive, year-round oceanographic study  also implements a
dedicated survey program, is still urgently needed as a complement to the ongoing sea ice monitoring. This would allow us
to investigate in more detail the outflow of Ice Shelf Water and the complex processes involved in the redistribution
of platelet crystals that emerge from the ice shelf cavity. Comparative analyses to other study regions are not possible at this
time, since, to our knowledge, no comparable long-term transects were carried out so far in other Antarctic fast-ice regions
with platelet layers beneath.
Examining the spatial distribution of snow over the bay, the considerably lower snow depth at ATKA24 compared to all
other sampling sites is striking Figure 4), and most likely related to the proximity to the ice-shelf edge in approximately 1
km distance. Due to the prevailing easterly winds in the bay (Figure 3), an east-west gradient in snow depth could have been
expected over the rest of the bay. However, this gradient cannot be determined on average over the entire time series. This is
mainly due to temporary local disturbance factors in the bay, such as icebergs and pressure ridges, which locally dominate
the snow distribution and thus lead to a comparatively homogeneous distribution of snow depth over the central part of Atka
Bay. A south-north survey across the bay at the beginning of austral summer 2018, however, revealed a trend of decreasing
snow depth towards the northern fast-ice edge, with a stronger gradient approx. 5 km from the ice edge (Figure 7), which is
in line with the northern boundary of the ice-shelf edge (Figure 1) and can therefore be explained by associated decreasing
offshore winds and consequently less snow redistribution.

Commented [TJ19]: Process not clearly demonstrated: why would offshore winds decrease at the ice shelf edge?.. less "channeling"?..

Commented [TJ20]: See previous comment. I don't think you discuss or show any specific snow redistribution process, unless it has escaped me…

[revised manuscript text omitted]

Hoppman et al. 2013.. not a complete reference.. what is it? A report?..Field summary?.. Need to be completed accordingly..

---

## Author Response (AR2)

Response to „**Review by Editor from 28 June 2020**"

Dear Jean-Louis,

We highly appreciate the great and intense work you put into revising our manuscript as your critical questions and remarks throughout the review process considerably improved our manuscript.

Regarding the given comments in the manuscript, we adjusted the following in the manuscript:

First of all, we implemented all suggested language/grammatical changes.

Comment 1: We agree that the term "snow redistribution" promises more than the manuscript is providing. We therefore followed the suggestion to change that towards "snow distribution" throughout the manuscript.

Comment 2: We added the reference Dieckmann et al. (1986).

Comment 3: We agree that this sentence was misleading and therefore rephrased it to
Atka Bay is seasonally sea-ice covered, and the water depth ranges between 80 m and 275 m with maximum depth in the central bay (Kipfstuhl, 1991).

Comment 4: We deleted the duplicate reference.

Comment 5: We agree that the described missing melting is related to surface melting only. The sentence is changed towards:
In the following summer, the ice does not disappear by surface melting in-situ, but breaks up and drifts out of the bay once the conditions are sufficiently unstable.

Comment 6/7: The given time period of 2010/11 to 2018/19 represents the time span of the continuous measurements in Atka Bay. This is why we focus our study on that period and highlight during that time the exceptional years 2012 and 2014.

Comment 8: We added the meant direction and changed therefore the sentence towards:
At each sampling site, up to five measurements are taken in an undisturbed area, one as the center measurement and four more at a distance of approx. 5 meters in each direction (north, east, south, and west), in order to account for the spatial variability of sea-ice and snow properties.

Comment 9: We agree that for these calculations, cases of dry and flooded snow need to be distinguished. We therefore add a second equation, taking also the latter case into account:
As soon as F gets negative, a flooding of the snow/ice interface is assumed, and with that the formation of snow ice. As the latter is assumed to have the same density as sea ice (Knight, 1988), freeboard is calculated for the flooded case as

$$-F = S - \frac{I + S - P\left(\frac{\rho_P - \rho_W}{\rho_W - \rho_I}\right)}{1 + \frac{\rho_S}{\rho_W - \rho_I}}$$

Comment 10: Thanks for that comment; we actually calculated the density of the platelet ice layer as suggested by you but phrased it in the manuscript in a "misunderstanding" way. We therefore rephrased it to:
The platelet-layer density $\rho_P$ is calculated by means of the platelet-layer ice volume fraction β as $\rho_P = \beta \cdot \rho_I + (1 - \beta) \cdot \rho_W$.

Comment 11: We rephrased the sentence towards:
Colors indicate the relative frequency of each shown wind direction to wind speed pair.

Comment 12: Due to the adjusted calculation, the average value between measured and calculated freeboard values is "negative" zero (-0,0045 m). The new statistics add up to:
According to Equation 1.1, 66% of the calculated freeboard values are smaller than the measured values. The difference between measured and calculated freeboard values ranges from -0.54 to 1.11 m with an average of $0.00 \pm 0.19$ m. Neglecting the underlying buoyant platelet layer in the calculation reduces the freeboard by $0.07 \pm 0.15$ m, whereas neglecting the snow layer on top of the sea ice increases the freeboard by $0.19 \pm 0.29$ m (Figure 5).

Comment 13: We agree and changed the gray scale towards the same as in Figure 4. However, we stick to the setting of the figure itself for consistency with the previous figure. Also, there is platelet ice visible, e.g., at ATKA11 in 2012.

Comment 14: The decreasing snow depth towards the ice shelf might be related to both, the easterly winds and the edge effect as shown in the Discussions.

Comment 15: We used consistently R throughout the manuscript.

Comment 16: We do not have an extensive data set on snow quality/properties in Atka Bay and can therefore not provide a proper comparison between the pack ice and fast ice area, respectively.

Comment 17: We agree and changed the sentence towards:

It is particularly remarkable that the average annual maximum platelet-layer thickness of 4 m (Table 1) is consistent with an earlier investigation at Atka Bay […]

Comment 18: We agree and added the ice rumples and ice rise also in our Figure 1c and do reference it instead of Hoppmann et al. (2015).

Comment 19/20: We agree that the way, the sentence is written, is misleading. We therefore rephrased it to:
A south-north survey across the bay at the beginning of austral summer 2018, however, revealed a trend of decreasing snow depth towards the northern fast-ice edge, with a stronger gradient approx. 5 km from the ice edge (Figure 7), which is in line with the northern boundary of the ice-shelf edge (Figure 1) and can therefore be explained by associated decreasing effects of the prevailing offshore winds. Consequently, also the measured snow depth is less in that part of the bay.

Comment 21: We agree that "linear" is kind of misleading in the given context. We therefore rephrased it to:
In contrast, the evolution of the second-year fast-ice thickness shows a different pattern.

Comment 22: We do consider Atka Bay as "key region" of the Southern Ocean as here the interactions and feedback mechanisms can not only be studied between atmosphere, sea ice and ocean, but also includes governing interactions to the neighboring ice shelfs. Furthermore, the Weddell Sea is one of the key regions of the global deep-water formation and contains at the same time relatively warm subsurface waters which are predicted to eventually push under the Filchner Ronne Iceshelf in the near future as stated in Vernet et al. (2019).

Comment 23: We agree (as mentioned in Comment 1) that Snow redistribution might be too ambiguous in the given context. We therefore rephrased it also here towards snow distribution only.

Comment 24: We feel super sorry that we missed to add you in the acknowledgements, Jean-Louis. Of course, we changed that immediately (and owe you a beer at the next opportunity!) ☺

[revised manuscript text omitted]

---

## Editor Decision (ED2)

Hi Sandra and co-authors, please consider the few remaining problems here below:

Comment 9: We agree that for these calculations, cases of dry and flooded snow need to be distinguished. We therefore add a second equation, taking also the latter case into account:

As soon as F gets negative, a flooding of the snow/ice interface is assumed, and with that the formation of snow ice. As the latter is assumed to have the same density as sea ice (Knight, 1988), freeboard is calculated for the flooded case as

$$-F = S - \frac{I + S - P\left(\frac{\rho_P - \rho_W}{\rho_W - \rho_I}\right)}{1 + \frac{\rho_S}{\rho_W - \rho_I}}$$

Not yet there, I believe…

a) I am not sure that, at this period of the year, flooded snow automatically turns into snow ice
b) If it is the case though, automatically the freeboard should be zero! I would have thought you would have considered "wet snow" (similar to loose platelet, in a way, but with different density depending on porosity).. floating snow, so to say…
c) I must admit I don't see how you reach that equation 1.2?...Please provide development, supposing you find it still valid, given b)…Also, in that case, what do you use for S: the thickness of observed dry snow?...
d) Have you taken the potential changes for negative freeboard into account in your figure 5?
e) Maybe we are "cutting hair in four", and that would not make any significant difference in the end(?)…  but it does make a difference in your comparison to measurements, obviously with your equation 1.2 (?) … you just need to be clear in the manuscript…(and in my mind :0))

Comment 11: We rephrased the sentence towards:
Colors indicate the relative frequency of each shown wind direction to wind speed pair.

I still have problem with that figure 3. To me it shows the range of wind velocities from various directions. It does not show "prevalence" of directions with regards to each others, is it?..., which is the main use you are making of it in the text, correct?.. and what is the use of that relative frequency (always extremely low %!) not used at all in the text, I believe..

Comment 12: Due to the adjusted calculation, the average value between measured and calculated freeboard values is "negative" zero (-0,0045 m). The new statistics add up to:
According to Equation 1.1, 66% of the calculated freeboard values are smaller than the measured values. The difference between measured and calculated freeboard values ranges from -0.54 to 1.11 m with an average of 0.00 ± 0.19 m. Neglecting the underlying buoyant platelet layer in the calculation reduces the freeboard by 0.07 ± 0.15 m, whereas neglecting the snow layer on top of the sea ice increases the freeboard by 0.19 ± 0.29 m (Figure 5).

So using your equation 1.2 changes things in the comparison to measurements… with now a "nul" average difference :0) …question is, is equation 1.2 valid?.. Also, how come these number change, and not the estimates neglecting either platelet or snow?...they should have changed too, since the calculation has changed? I must admit I am lost!..

Comment 14: The decreasing snow depth towards the ice shelf might be related to both, the easterly winds and the edge effect as shown in the Discussions.

I know, but what I am saying is that the "eastern edge" is not different from the "western edge" if we look at the figures.. both edges show a decreasing snow thickness.. so you cannot say what you say there..

---

## Author Response (AR3)

**Response to „Review by Editor from 07 July 2020"**

Dear Jean-Louis, thanks a lot (again) for your critical comments and questions.

Please find our comments to your latest review below.

**The "flooding issue"**

We very much appreciate your constructive criticism to the calculation of the negative freeboard case. We must admit that mistakes were introduced into our calculation and we decided to start from scratch again – specially to clean up the confusing indices. In addition, we separated now the density of the soaked snow. Since this layer is a mixture of water, snow, ice and air bubbles, its density is rather difficult to determine accurately. After a more detailed literature research, we set the density of the soaked snow to $\rho_{swet}= 920$ kg m$^{-3}$.

Thus, the calculation corresponds to:

$$\rho_W \cdot (I_{draft} + P) = \rho_i \cdot I + \rho_{swet} \cdot S_{wet} + \rho_S \cdot S_{dry} + \rho_p \cdot P \qquad\qquad I_{draft} = I + S_{wet}$$

$$\rho_W \cdot (I + S_{wet} + P) = \rho_i \cdot I + \rho_{slush} \cdot S_{wet} + \rho_S \cdot S_{dry} + \rho_p \cdot P$$

$$S_{wet} = \frac{I \cdot (\rho_i - \rho_W) + \rho_S \cdot S_{dry} + P \cdot (\rho_P - \rho_W)}{\rho_W - \rho_{swet}}$$

$S_{dry}$ represents here the measured snow depth S and $S_{wet}$ the negative freeboard, leading to:

$$F = -\frac{I \cdot (\rho_i - \rho_W) + \rho_S \cdot S + P \cdot (\rho_P - \rho_W)}{\rho_W - \rho_{swet}}$$

Based on that, we rephrased the paragraph to:

"As soon as F becomes negative, the involved components of the above-mentioned hydrostatic equilibrium are assumed to be balanced after the flooding of the snow/ice interface. Here, the depth of the wet soaked snow is considered as equal to the absolute value of the freeboard. Thus, the latter is calculated for the flooded case as

$$F = -\frac{I \cdot (\rho_i - \rho_W) + \rho_S \cdot S + P \cdot (\rho_P - \rho_W)}{\rho_W - \rho_{swet}}. \qquad\qquad\qquad\qquad\qquad\qquad \text{(Eq. 1.2)}$$

In Equation 1.1 and 1.2, I refers to sea-ice thickness, S to the dry snow depth, P to platelet-layer thickness, the indices I refers to sea ice, S to dry snow, P to the platelet layer, and W to water. Constant typical densities of $\rho_W = 1032.3$ kg m$^{-3}$, $\rho_S = 330$ kg m$^{-3}$ and $\rho_I = 925$ kg m$^{-3}$ are assumed in this study. The density of the soaked snow, which is a mixture of water, snow, ice and air bubbles, is here assumed as $\rho_{swet}= 920$ kg m$^{-3}$ (Wang et al., 2015). "

The related paragraph and figure in Section 3.1 were adapted to:

"According to Equation 1.1, 70% of the calculated freeboard values are smaller than the measured values. The difference between measured and calculated freeboard values ranges from -0.54 to 1.26 m with an average of -0.02 ± 0.18 m. Neglecting the underlying buoyant platelet layer in the calculation reduces the freeboard by 0.03 ± 0.17 m, whereas neglecting the snow layer on top of the sea ice increases the freeboard by 0.17 ± 0.25 m (Figure 5)."

[Figure]

**The "wind velocity issue"**

Thanks a lot for pointing out the "Relative frequency in %" confusion – this was changed to read "Relative frequency" only, meaning e.g. 0.5 equals 50%.

The figure shows both points you mentioned: The range of wind velocities from various directions (x/y axis) and the relative occurrences of each wind direction/wind speed pair. And this is summarized in the manuscript as

"[…] This leads to prevailing persistent and strong easterly winds which exhibit a seasonal cycle with strongest winds during winter time (Figure 3). […]".

**The "edge effect issue"**

Thanks for clarifying your previous Comment 14. We agree and adjusted the statement to read:

[revised manuscript text omitted]